# Analysis of common differential gene expression in synovial cells of osteoarthritis and rheumatoid arthritis

**Chang-sheng Liao**[1,2], **Fang-zheng He**[1,2], **Xi-yong Li**[1,2], **Yan Zhang**[1,2], **Peng-fei Han**[1] *

**1** Department of Orthopaedics, Heping Hospital Affiliated to Changzhi Medical College, Changzhi, P.R. China, **2** Department of Graduate School, Graduate Student Department of Changzhi Medical College, Changzhi, P.R. China

* 18003551149@163.com

## Abstract

### Objective

To elucidate potential molecular mechanisms differentiating osteoarthritis (OA) and rheumatoid arthritis (RA) through a bioinformatics analysis of differentially expressed genes (DEGs) in patient synovial cells, aiming to provide new insights for clinical treatment strategies.

### Materials and methods

Gene expression datasets GSE1919, GSE82107, and GSE77298 were downloaded from the Gene Expression Omnibus (GEO) database to serve as the training groups, with GSE55235 being used as the validation dataset. The OA and RA data from the GSE1919 dataset were merged with the standardized data from GSE82107 and GSE77298, followed by batch effect removal to obtain the merged datasets of differential expressed genes (DEGs) for OA and RA. Intersection analysis was conducted on the DEGs between the two conditions to identify commonly upregulated and downregulated DEGs. Enrichment analysis was then performed on these common co-expressed DEGs, and a protein-protein interaction (PPI) network was constructed to identify hub genes. These hub genes were further analyzed using the GENEMANIA online platform and subjected to enrichment analysis. Subsequent validation analysis was conducted using the GSE55235 dataset.

### Results

The analysis of differentially expressed genes in the synovial cells from patients with Osteoarthritis (OA) and Rheumatoid Arthritis (RA), compared to a control group (individuals without OA or RA), revealed significant changes in gene expression patterns. Specifically, the genes APOD, FASN, and SCD were observed to have lower expression levels in the synovial cells of both OA and RA patients, indicating downregulation within the pathological context of these diseases. In contrast, the SDC1 gene was found to be upregulated, displaying higher expression levels in the synovial cells of OA and RA patients compared to normal

**Data Availability Statement:** All relevant data are within the manuscript and its Supporting Information files.

**Funding:** The author(s) received no specific funding for this work.

**Competing interests:** The authors have declared that no competing interests exist.

controls.Additionally, a noteworthy observation was the downregulation of the transcription factor PPARG in the synovial cells of patients with OA and RA. The decrease in expression levels of PPARG further validates the alteration in lipid metabolism and inflammatory processes associated with the pathogenesis of OA and RA. These findings underscore the significance of these genes and the transcription factor not only as biomarkers for differential diagnosis between OA and RA but also as potential targets for therapeutic interventions aimed at modulating their expression to counteract disease progression.

## Conclusion

The outcomes of this investigation reveal the existence of potentially shared molecular mechanisms within Osteoarthritis (OA) and Rheumatoid Arthritis (RA). The identification of APOD, FASN, SDC1, TNFSF11 as key target genes, along with their downstream transcription factor PPARG, highlights common potential factors implicated in both diseases. A deeper examination and exploration of these findings could pave the way for new candidate targets and directions in therapeutic research aimed at treating both OA and RA. This study underscores the significance of leveraging bioinformatics approaches to unravel complex disease mechanisms, offering a promising avenue for the development of more effective and targeted treatments.

## Introduction

As the most common inflammatory joint diseases globally [1, 2], osteoarthritis (OA) and rheumatoid arthritis (RA) mainly manifest as joint pain, swelling, and stiffness [3, 4]. Their high incidence and high disability rates not only bring heavy physical and mental burdens to patients but also impose tremendous pressure on social medical resources [5]. Although researchers have made significant progress in multiple areas of OA and RA, such as molecular mechanisms, early diagnosis, targeted therapy, the application of biological agents, and the rise of regenerative medicine over the past few decades [6–8], there are still many shortcomings. It is particularly noteworthy that OA and RA often intertwine clinically, with OA possibly secondary to RA or occurring simultaneously with RA [9, 10]. Therefore, separating the two during the diagnosis and treatment of OA and RA may overlook their interactions and connections. Joint research on OA and RA is thus particularly important, as it will help improve diagnostic accuracy, optimize treatment plans, and provide patients with more effective treatment methods.

In recent years, studies have revealed significant similarities between OA and RA in terms of inflammatory responses, immune regulation, and cellular signal transduction [11–13], prompting researchers to re-examine the connection between the two diseases and explore similar molecular mechanisms and common therapeutic targets. With the development of bioinformatics, it has provided powerful tools for disease research [14]. The purpose of this study is to use bioinformatics methods to conduct a dual-disease joint study of osteoarthritis and rheumatoid arthritis, revealing the internal connections and differences between the two, identifying new biomarkers and potential disease-related genes, deepening our understanding of OA and RA, and providing strong support for the development of new treatment methods, diagnostic tools, and prevention strategies, especially achieving greater breakthroughs in combination therapy.

In this study, we used differential expression analysis methods based on RNA sequencing data and transcription factor prediction algorithms to screen for differentially expressed genes and transcription factors shared between RA and OA. This approach aims to deeply explore the common characteristics of these two diseases at the molecular level, contribute a new perspective to basic scientific research, and provide valuable clues for uncovering the pathogenesis of both diseases, potentially having a positive impact on clinical medicine. Finally, we also hope that these findings can provide references for the early prevention, diagnosis, and treatment strategy formulation of OA and RA.

## I. Materials and methods

### 1. Data acquisition

In this study, we primarily utilized microarray technology for gene expression analysis, acquiring datasets related to osteoarthritis (OA) and rheumatoid arthritis (RA) from the public Gene Expression Omnibus (GEO) database. We searched and extracted datasets from the GEO database using the following keywords: "osteoarthritis," "rheumatoid arthritis," "microarray," "human samples," and the corresponding disease-specific gene expression patterns. The datasets included in our analysis were GSE82107 (OA) and GSE77298 (RA), along with GSE55235 and GSE1919, which encompass data for both OA and RA. The GSE82107 (OA) dataset comprised 17 samples, including 10 disease samples and 7 control samples; GSE77298 (RA) included 23 samples, with 16 disease samples and 7 control samples; GSE55235 contained 30 samples, with 10 OA samples, 10 RA samples, and 10 control samples; GSE1919 comprised 15 samples, with 5 OA samples, 5 RA samples, and 5 control samples. Notably, the GSE82107 (OA) and GSE77298 (RA) datasets originated from the same laboratory in the Netherlands, while GSE55235 and GSE1919 were collected by the same laboratory in Germany. Regarding age groups and gender information, due to limitations during the data collection process, detailed age group and gender information for all samples could not be obtained. Although the age and gender information of samples are important background variables for some analyses, our primary goal was to identify differentially expressed genes associated with osteoarthritis and rheumatoid arthritis and to explore how these genes function in the progression of these diseases. Therefore, we decided to include the aforementioned datasets in our study.

### 2. Data preprocessing and identification of differentially expressed genes

Initially, according to the characteristics of each dataset, GSE82107 was organized for OA data, GSE77298 for RA, whereas GSE1919 included samples for both OA and RA, hence it was divided into two annotated expression datasets: GSE1919 (OA) and GSE1919 (RA). To ensure the accuracy of the analysis, rows (genes) or columns (samples) with a large number of missing values were removed to improve data quality. For genes appearing multiple times within the dataset, their expression levels were averaged to simplify the analysis process and minimize redundancy. Subsequently, the processed data were normalized to eliminate systematic bias between experiments and ensure comparability across datasets. In the final stage of this phase, we employed a differential expression gene (DEGs) analysis package for conducting differential expression analysis across the four annotated expression datasets (GSE82107 for OA, GSE77298 for RA, and GSE1919 divided into OA and RA categories), aiming to compare the gene expression differences between the normal control group and the OA and RA groups. The analysis tool selected was DESeq2, a widely used and validated R package for count data differential expression analysis. DESeq2 fits the gene expression data using a negative binomial distribution and employs the Wald test to assess differences between gene expression levels. We set a stringent criterion for DEGs identification: an adjusted P-value (adjP) less than 0.05

and an absolute log2 fold change (|log2FC|) of at least 1. These criteria were designed to ensure that the identified DEGs showed significant statistical differences and meaningful biological variation in expression levels. Through this rigorous filtering process, our goal was to select those genes most likely to play a crucial role in the pathology of OA and RA from a vast pool of candidates.

### 3. Batch effect correction and differential analysis

In order to mitigate the batch effects that may arise from different experimental platforms, this study employed a batch correction algorithm. We merged pairs of normalized expression datasets from GSE82107 (OA), GSE1919 (OA), GSE1919 (RA), and GSE77298 (RA) and utilized the ComBat function provided by the sva package in RStudio for batch effect adjustment. Following the correction, we conducted differential expression gene analysis using the limma package to identify statistically significant differentially expressed genes within the consolidated dataset.

### 4. Selection of common differentially expressed genes between OA and RA

After batch effect correction and differential expression analysis, DEGs from the OA and RA datasets were identified. An intersection analysis of the DEG sets from both diseases allowed us to successfully identify common differentially expressed genes, including both upregulated and downregulated genes.

### 5. Enrichment analysis

To comprehensively elucidate the biological functions and pathways associated with the identified common differentially expressed genes (DEGs), we employed the clusterProfiler package in R for an integrated analysis. This encompassed Gene Ontology (GO) enrichment analyses for Biological Process (BP), Cellular Component (CC), and Molecular Function (MF), as well as Kyoto Encyclopedia of Genes and Genomes (KEGG) pathway analyses for metabolism and signal transduction. The GO analysis facilitated the understanding of the functional attributes of individual genes or proteins, while the KEGG analysis revealed their roles within broader biological networks. By integrating these two analyses, we were able to comprehend the changes in gene expression from multiple perspectives, thereby providing deeper insights into biological processes and generating new hypotheses and directions for future research. In both analyses, we set a significance threshold of a P-value less than 0.05 to ensure that the selected categories and pathways were statistically significant.

### 6. Construction and analysis of the protein-protein interaction (PPI) network and selection of hub genes

Utilizing the STRING online platform, we constructed a human protein-protein interaction network incorporating 30 common differentially expressed genes (DEGs) from both osteoarthritis (OA) and rheumatoid arthritis (RA), focusing on association data for H. sapiens. To ensure the credibility of the protein interactions within the network, we established a minimum interaction score threshold of 0.400. Subsequently, the network was analyzed using the CytoHubba plugin within Cytoscape software (version 3.10.1) to identify hub genes. This analysis employed four distinct centrality algorithms—Degree, Closeness Centrality, Betweenness Centrality, and Modularity Class (MCC)—to effectively detect genes playing central roles in the network. This multifaceted approach allowed us to identify hub genes with significant impacts on the network's structure and function, circumventing the limitations of relying on a

single network characteristic for gene identification and ensuring that the identified hub genes are validated from multiple perspectives for their importance within the network.

## 7. GENEMANIA online analysis

For further investigation into the interactions and functional associations between the ten selected hub genes from this study, the advanced gene function prediction tool GENEMANIA was used. By entering the names of these ten hub genes on the GENEMANIA website and selecting the appropriate species, "Homo sapiens", an interaction network map was automatically constructed based on their known interactions and functional associations.

## 8. Enrichment analysis of hub genes

Utilizing the clusterProfiler package in R, we conducted an integrated Gene Ontology (GO) and Kyoto Encyclopedia of Genes and Genomes (KEGG) pathway enrichment analysis for the selected hub genes. This analysis was designed to precisely elucidate the roles of hub genes in specific biological pathways and functions, to substantiate their biological significance, and to construct a gene interaction network for a deeper understanding of their synergistic actions at the systems level. Throughout the analysis, we established a p-value threshold of less than 0.05 to ensure that the identified biological processes, cellular components, molecular functions, and pathways hold statistical significance, thereby assisting us in filtering out the genes and pathways most relevant and likely to impact biological states.

## 9. Independent validation of DEGs using an additional dataset

For the validation phase, we utilized the publicly available GSE55235 dataset, which was stratified based on clinical data into three distinct groups: normal, osteoarthritis (OA), and rheumatoid arthritis (RA). To streamline the analysis, we derived two comparative sub-datasets: GSE55235OA, which included samples from the normal and OA groups, and GSE55235RA, which included samples from the normal and RA groups. Prior to conducting differential expression analysis, we performed essential preprocessing steps, including normalization, to minimize technical variability and ensure data comparability. The differential expression analysis aimed to identify genes that were significantly differentially expressed (DEGs) between the normal group and each of the disease-specific groups (OA and RA). Subsequently, we specifically examined the overlap between the DEGs identified in this dataset and the ten hub genes previously determined from our primary analysis. To be considered validated, a gene had to exhibit a statistically significant difference in expression (p-value < 0.01) when comparing the normal group to either the OA or RA group.

## 10. Transcription factor differential analysis

The selected 10 hub genes were analyzed in TRRUST (Transcriptional Regulatory Relationships Unraveled by Sentence-based Text mining), a database dedicated to the interactions between mammalian transcription factors and their target genes. Setting the species to human, we searched for associated transcription factors of these 10 hub genes within the TRRUST database, obtaining a list of transcription factors related to the 10 hub genes. These results, together with the batch-effect corrected OA and RA annotated expression dataset's transcriptome data from the training group, were then subjected to a differential expression analysis for the candidate transcription factors using R. This analysis utilized packages to compare the differences in transcription factor expression levels between the batch-effect corrected combined OA and RA groups and the normal control group. Through comparative analysis, we

identified significant differences in the expression levels of related transcription factors between the OA and RA groups compared to the normal control group. These differences were visualized using violin plots, with asterisks indicating statistical significance (*** for p-value<0.001; ** for p-value<0.01; * for p-value<0.05; ns for not significant).

## 11. Statistical methods, software, and tools

All statistical analyses were conducted within the R environment. A two-sample t-test analyzed gene expression differences between groups. The Benjamini-Hochberg method was applied for multiple testing correction to control the false discovery rate (FDR). R language (version 4.3.1) was utilized for data processing and statistical analysis.

# II. Results

## 1. Identification of differentially expressed genes

Upon analysis, a series of differentially expressed genes (DEGs) were identified across various datasets. Specifically, the GSE1919 dataset revealed 175 DEGs in the osteoarthritis (OA) group and 452 DEGs in the rheumatoid arthritis (RA) group. The GSE82107 dataset identified 8 DEGs in the OA group, while the GSE77298 dataset disclosed 285 DEGs in the RA group (Figs 1–4).

## 2. Batch correction and differential analysis

After undergoing standardization procedures, we initially integrated the annotated expression datasets from the GSE1919 osteoarthritis (OA) group, GSE82107 OA group, GSE1919 rheumatoid arthritis (RA) group, and GSE77298 RA group. We then eliminated batch effects from the merged dataset. Subsequent differential expression analysis revealed 39 differentially expressed genes (DEGs) in the OA group and 519 DEGs in the RA group (Figs 5–12).

## 3. Selection of commonly expressed genes between OA and RA

We conducted a differential expression analysis on the merged and batch-effect-corrected datasets of the osteoarthritis (OABatch) and rheumatoid arthritis (RABatch) groups, yielding their respective differentially expressed genes. By intersecting the differentially expressed genes from both groups, we precisely identified 15 genes that were upregulated and 15 genes that were downregulated. The upregulated genes include STMN2, SDC1, PLXNC1, CDH11, IGJ, CXCL10, MARCKS, HK3, TNFSF11, IGHM, KDELR3, TNFRSF11A, CRIP1, ARHGAP5, and CD300C. The 15 downregulated genes are FASN, KLF9, ADH1A, ADIPOQ, FABP4, SHC3, APOD, TF, PRKAR2B, PLIN1, ADH1B, ZBTB16, SCD, SOX13, and P2RY14 (Figs 13 and 14).

## 4. Enrichment analysis

The enrichment analysis was conducted on the 30 commonly expressed differentially expressed genes (DEGs) using Gene Ontology (GO) and Kyoto Encyclopedia of Genes and Genomes (KEGG) with a selection criterion of "P < 0.05."The enrichment analysis revealed that the common DEGs were primarily enriched in the following aspects:Biological Pathway (BP): Major enrichment was observed in pathways related to myeloid cell differentiation, myeloid leukocyte differentiation, and temperature homeostasis. Cellular Component (CC): The genes were predominantly enriched in components associated with the external side of the plasma membrane. Molecular Function (MF): Significant enrichment was found in functions related to oxidoreductase activity, specifically acting on the CH-OH group of donors with

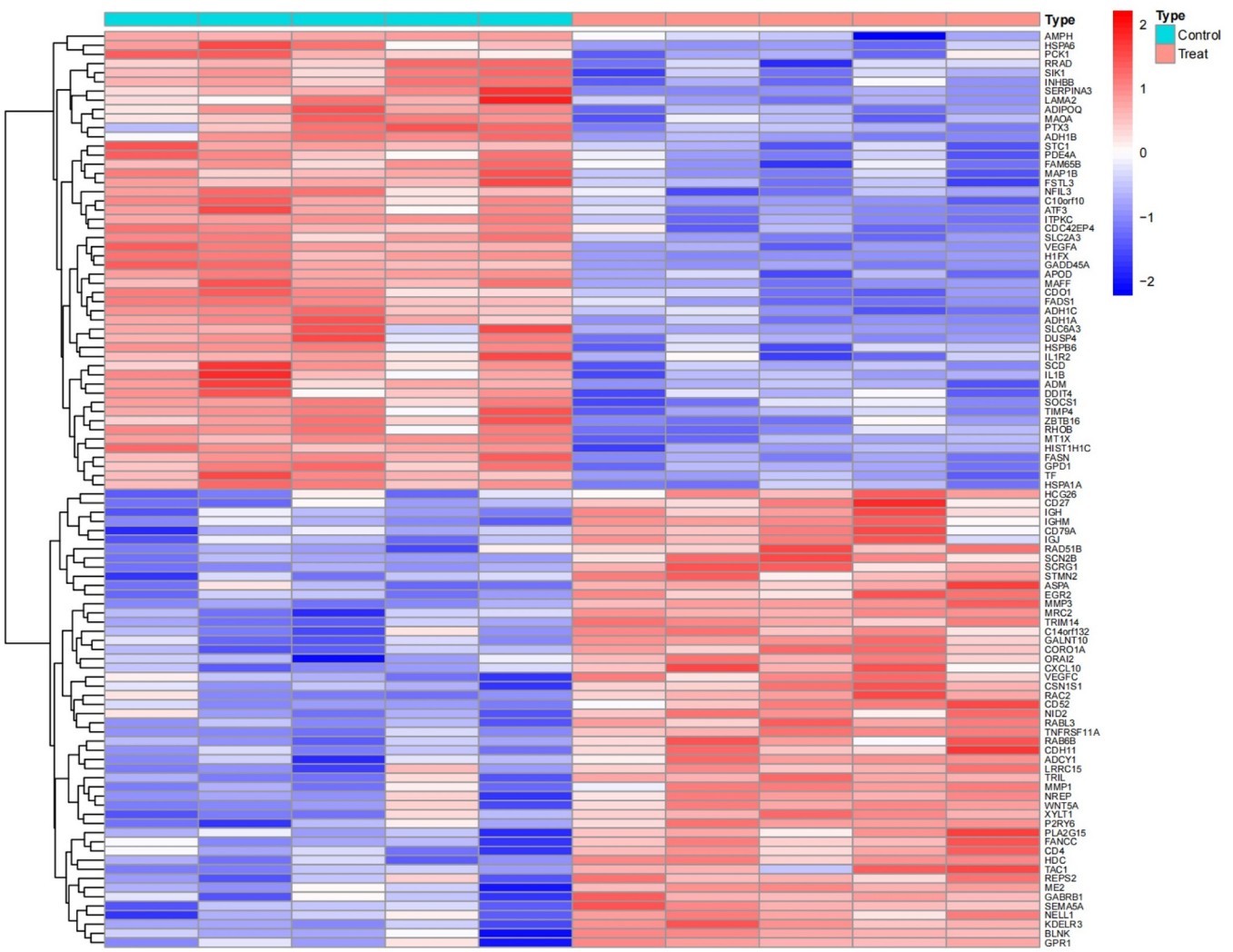

**Fig 1. Displays a heatmap of differentially expressed genes in the GSE1919 OA group.**

NAD or NADP as acceptors, and cytokine activity. KEGG Pathways: The genes showed significant enrichment in pathways including Alcoholic liver disease, PPAR signaling pathway, Insulin signaling pathway, and Glycolysis/Gluconeogenesis among others.These findings indicate that the shared DEGs between OA and RA are involved in critical biological processes and pathways that may underline the pathophysiological similarities between these two diseases. The enrichment in specific signaling pathways such as PPAR and Insulin signaling further suggests potential therapeutic targets and mechanisms for intervention(Figs 15–18).By integrating the results from both GO and KEGG analyses, we gain a more comprehensive understanding of the roles these shared DEGs play in disease pathology, providing novel insights for the development of targeted therapeutic strategies.

## 5. Construction of protein-protein interaction network (PPI) and selection of hub genes

In this study, a Protein-Protein Interaction (PPI) network was constructed based on Differentially Expressed Genes (DEGs) common to Osteoarthritis (OA) and Rheumatoid Arthritis

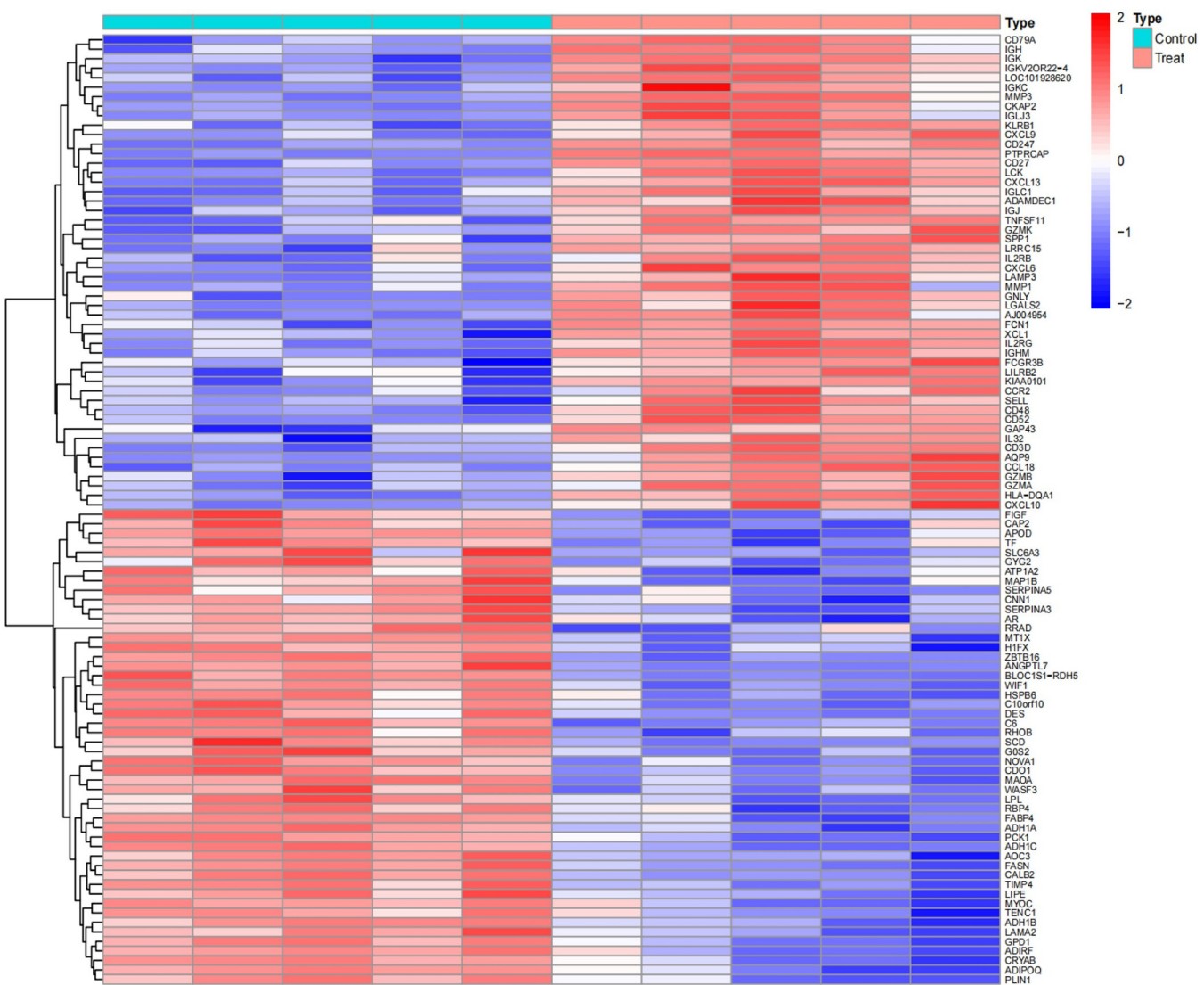

**Fig 2. Shows a heatmap for the GSE1919 RA group.**

(RA). As depicted in Fig 5A, the network incorporated 30 shared DEGs, represented by 34 nodes and 58 edges, where nodes denote proteins encoded by DEGs and edges indicate interactions between proteins. Utilizing Cytoscape software and various algorithms available in the cytohubba plugin, we further analyzed the network and identified ten key hub genes that met multiple selection criteria: PLIN1, SDC1, CXCL10, SCD, FABP4, FASN, TNFSF11, JCHAIN, APOD, and ADIPOQ. As shown in Fig 5B, the position of these hub genes within the network and the intensity of node color reflect their connectivity and significance in the network(Figs 19 and 20).

## 6. GENEMANIA online analysis

The analysis conducted on the GENEMANIA website for the 10 hub genes (DEGs) specifically PLIN1, SDC1, CXCL10, SCD, FABP4, FASN, TNFSF11, JCHAIN, APOD, and ADIPOQ revealed their roles in various biological processes. These genes are either directly involved or

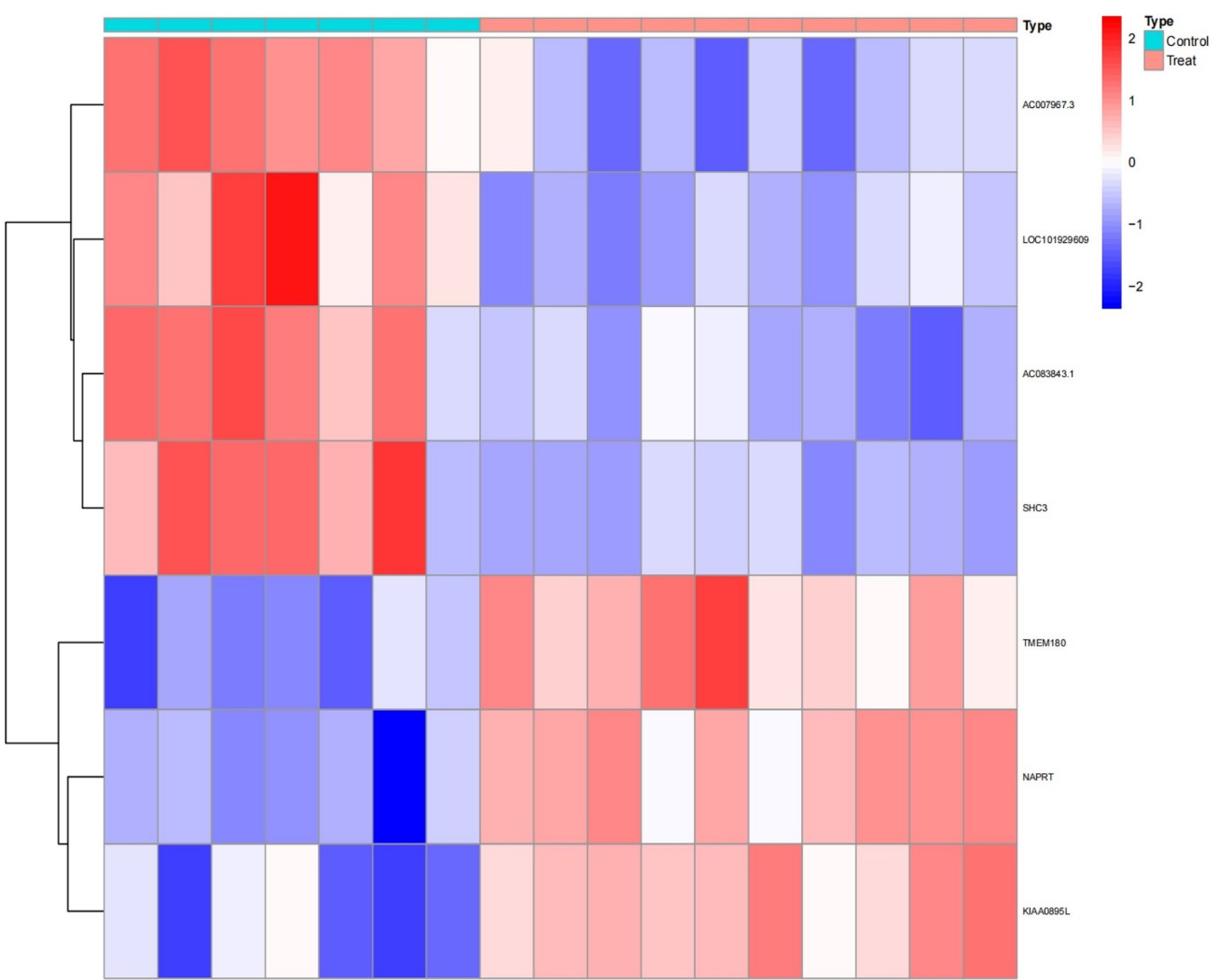

**Fig 3. Illustrates differentially expressed genes in the GSE82107 OA group.**

indirectly partake through interactions with other genes in crucial metabolic and thermogenic processes. The processes identified include:Temperature Homeostasis, Cold-Induced Thermogenesis, Regulation of Cold-Induced Thermogenesis, Adaptive Thermogenesis, Regulation of Lipid Metabolic Process, Diacylglycerol Metabolic Process, Fatty Acid Transport.This analysis underscores the importance of these hub genes in regulating metabolic and thermogenic processes, implicating potential mechanistic links between these processes and the pathophysiology of Osteoarthritis (OA) and Rheumatoid Arthritis (RA). The involvement of these genes in such a wide array of key biological processes highlights their potential as targets for therapeutic intervention, aiming to modulate underlying disease mechanisms and symptoms(Fig 21).

## 7. Enrichment analysis of hub genes

The hub genes identified earlier underwent Gene Ontology (GO) and Kyoto Encyclopedia of Genes and Genomes (KEGG) enrichment analysis, adopting "P<0.05" as the selection criterion. The results of the enrichment analysis illustrate the significant roles these hub genes play

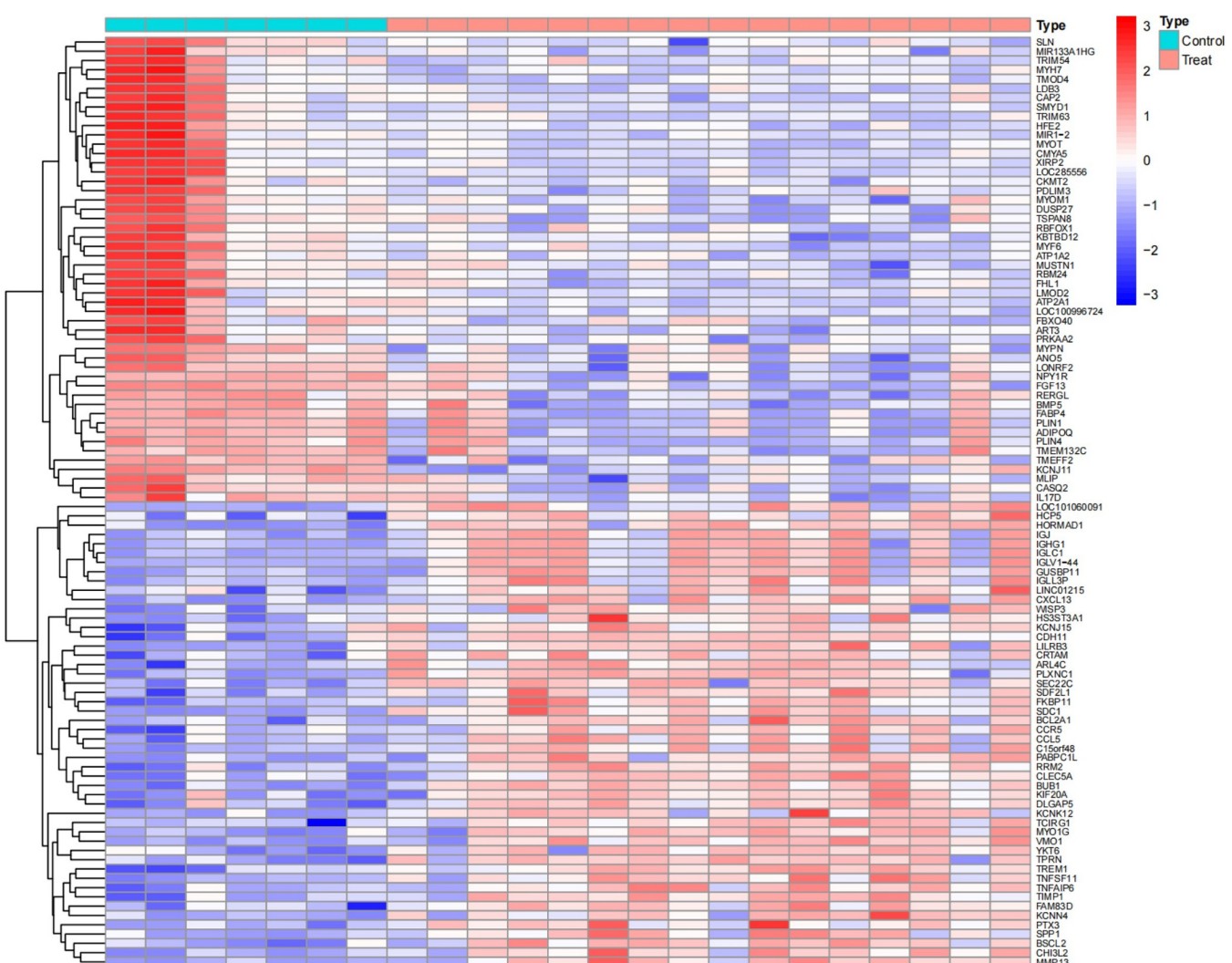

**Fig 4. Details the heatmap for the GSE77298 RA group.** In these heatmaps, red indicates upregulated genes, and blue represents downregulated genes. "Control" denotes the normal control group, while "Treat" refers to either the OA or RA group.

in various biological processes, cellular components, and molecular functions, as well as their involvement in essential signaling pathways.Biological Pathway (BP): The hub genes are predominantly enriched in processes critical for metabolic and thermogenic regulation, including temperature homeostasis, lipid transport, positive regulation of cold-induced thermogenesis, regulation of cold-induced thermogenesis, cold-induced thermogenesis, and adaptive thermogenesis. These pathways are indicative of the genes' roles in maintaining bodily temperature and energy homeostasis under varying environmental conditions. Cellular Component (CC): A significant enrichment was observed in lipid droplets, suggesting these genes are functionally connected to lipid storage and metabolism. The association with lipid droplets underscores the importance of lipid regulation in the physiological processes mentioned.Molecular Function (MF): Enrichment in cytokine activity and receptor ligand activity points to the essential roles these genes may play in cellular signaling and communication, particularly in regulating immune responses and interactions between cells.KEGG Pathways: Notably, the hub genes show significant enrichment in the PPAR signaling pathway and AMPK signaling

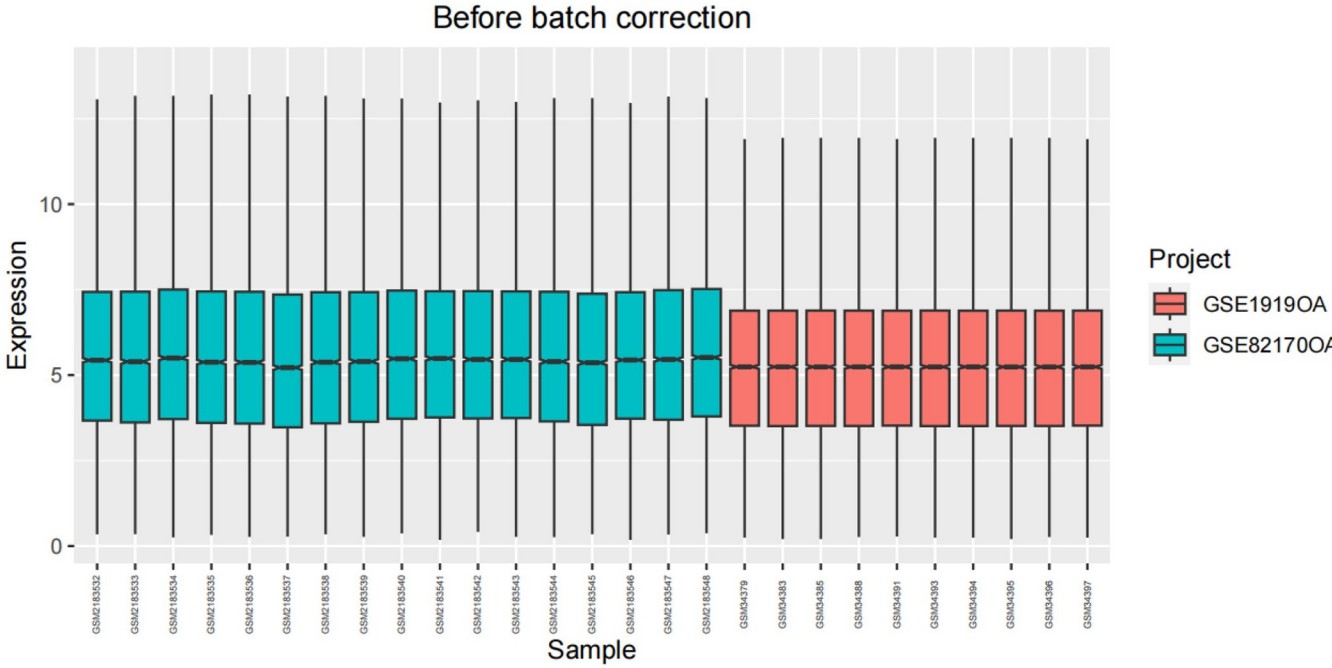

**Fig 5. Boxplot of the OA group before batch correction.**

pathway. The PPAR signaling pathway is known for its role in regulating lipid metabolism, adipocyte differentiation, and inflammation, while the AMPK signaling pathway is a crucial energy sensor that helps maintain cellular energy balance. These pathways are vital for metabolic health, including aspects related to the enriched biological processes earlier mentioned. This enrichment analysis highlights the intricate network of biological functions and pathways in which the hub genes are involved. Their roles in temperature regulation, lipid metabolism, and signaling pathways suggest potential mechanistic links to the pathophysiology of metabolic disorders and offer insights that could pave the way for therapeutic interventions targeting these molecular pathways(Figs 22 and 23).

## 8. Independent validation of DEGs using an additional dataset

In performing a validation analysis with a designated validation dataset, a p-value threshold of <0.01 was selected to determine significant differences. The analysis aimed to validate the differential expression patterns of specific genes of interest in the context of Osteoarthritis (OA) and Rheumatoid Arthritis (RA) compared to a normal control group.The results of the validation analysis revealed significant differences in the expression levels of APOD, FASN, SDC1, and TNFSF11 genes between the disease groups (OA and RA) and the normal control group. More specifically:APOD and FASN genes were found to be significantly downregulated in both OA and RA disease groups compared to the normal controls. This implies that these genes are less active in the pathology of both OA and RA, suggesting a potential protective or disease-modifying role when expressed at normal levels. Conversely, SDC1 and TNFSF11 genes showed significantly elevated levels of expression in the disease groups, indicating an upregulated state. The higher expression levels of these genes in the context of OA and RA highlight their possible involvement in disease progression or inflammatory processes.The validation analysis underscores the importance of these genes as potential biomarkers or

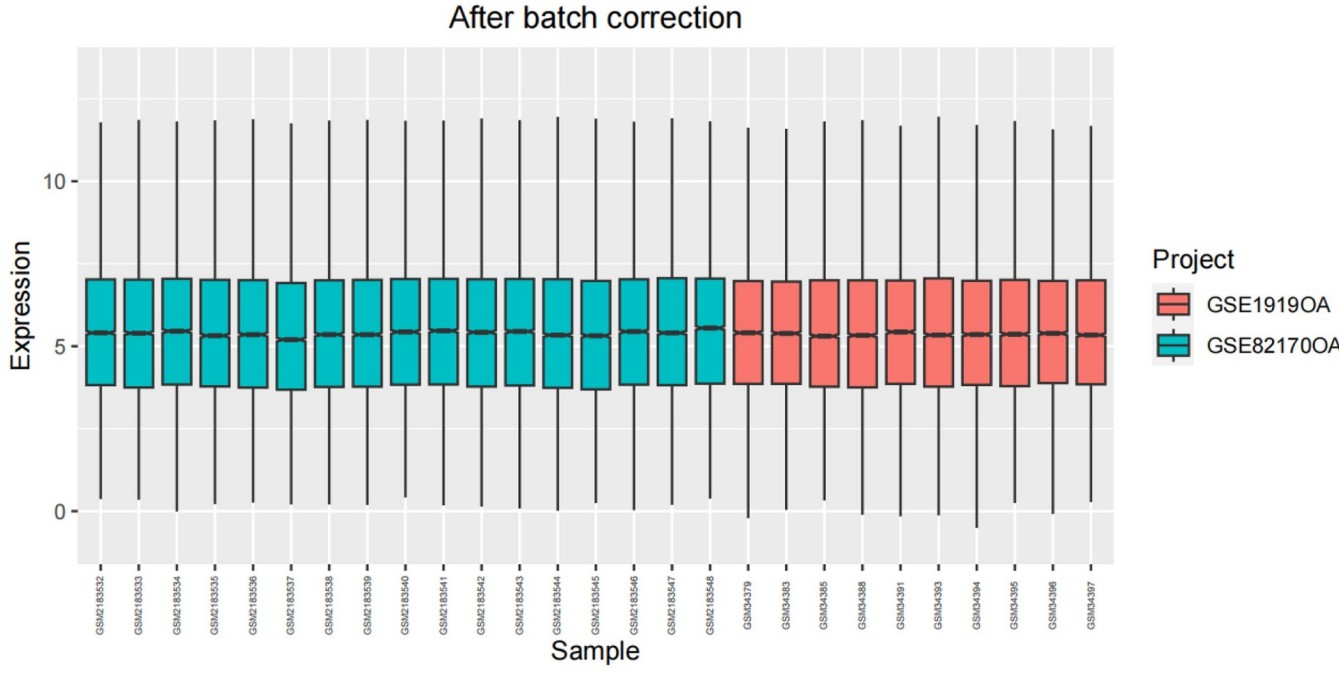

**Fig 6. Boxplot of the OA group after batch correction.**

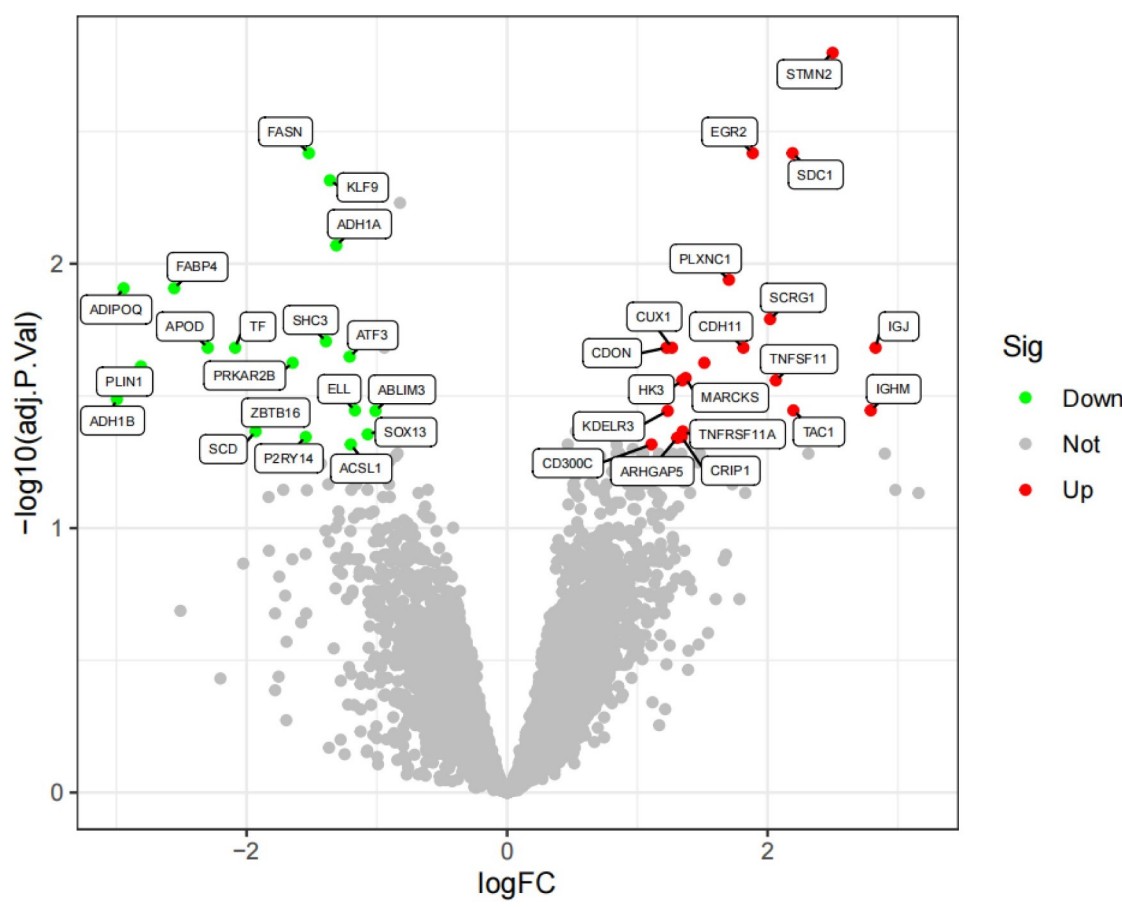

**Fig 7. Boxplot of the RA group before batch correction.**

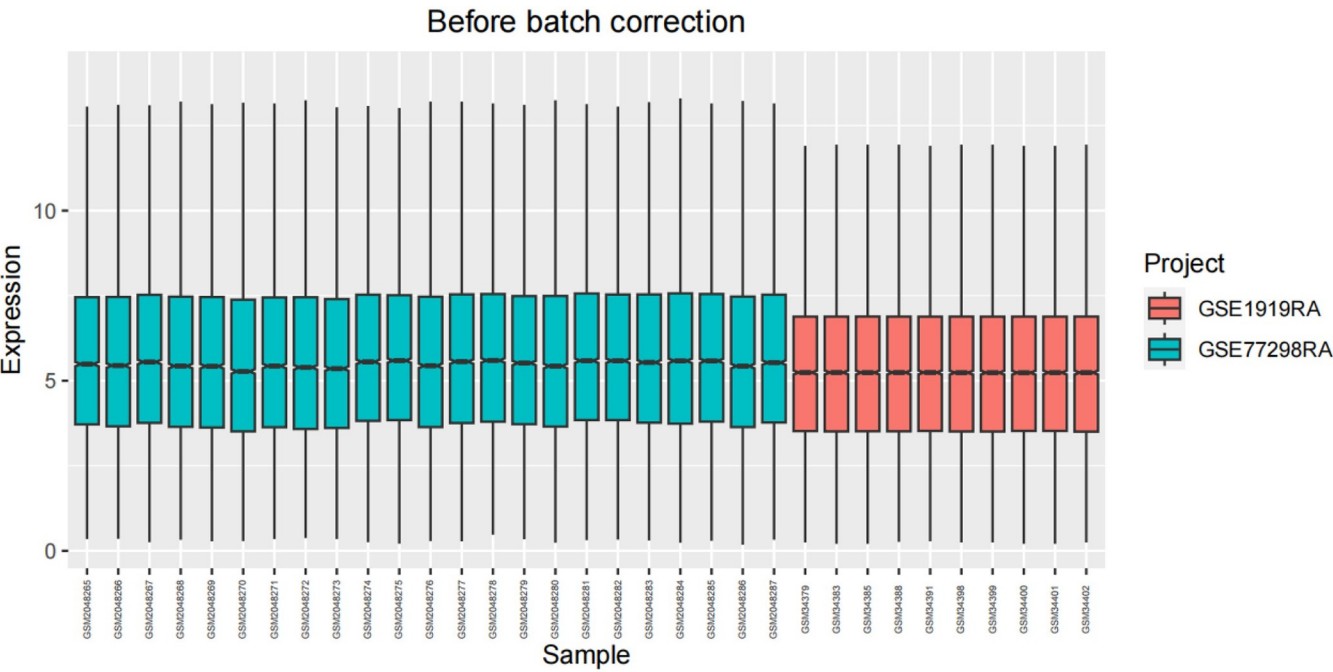

**Fig 8. Boxplot of the RA group after batch correction.**

**Fig 9. Volcano plot of the OA group, with green indicating low expression and red indicating high expression.**

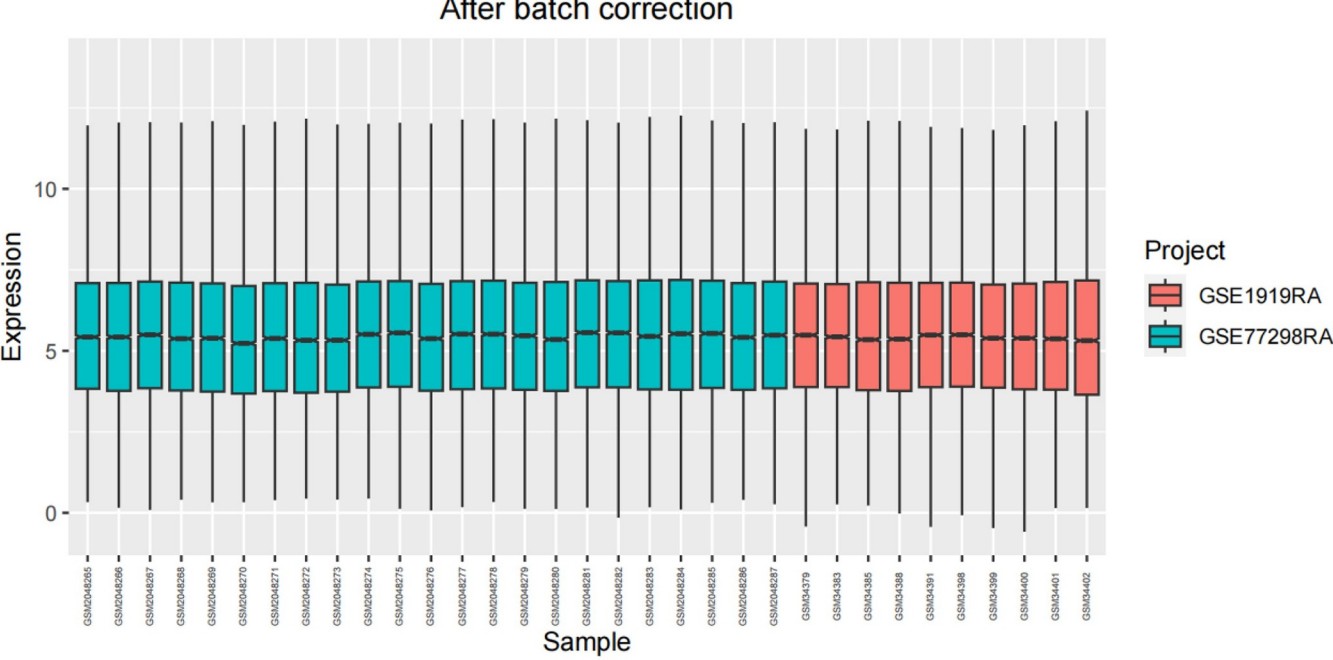

**Fig 10. Volcano plot of the RA group, with green indicating low expression and red indicating high expression.**

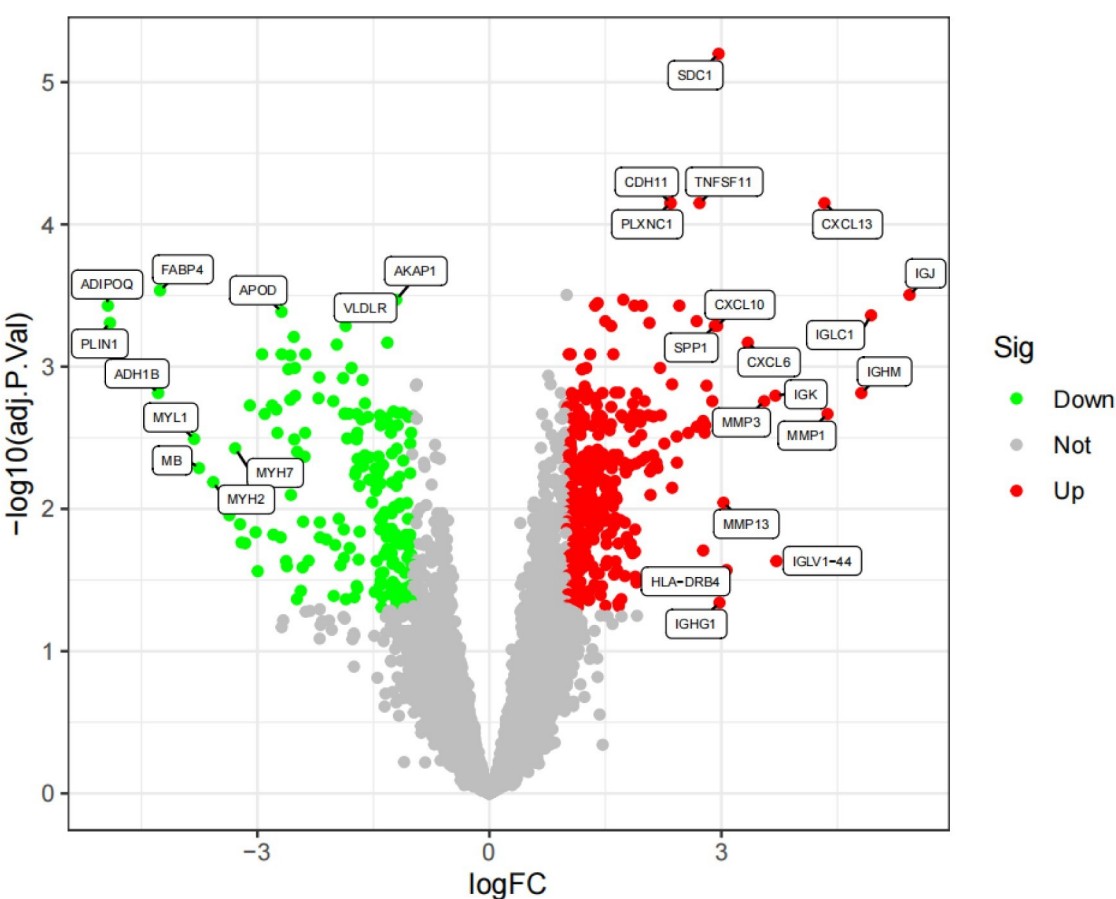

**Fig 11. Heatmap of the OA group, where light blue represents the normal control group, pink represents the disease group, and green and orange differentiate the distinct datasets included, with red signifying high expression and blue representing low expression.**

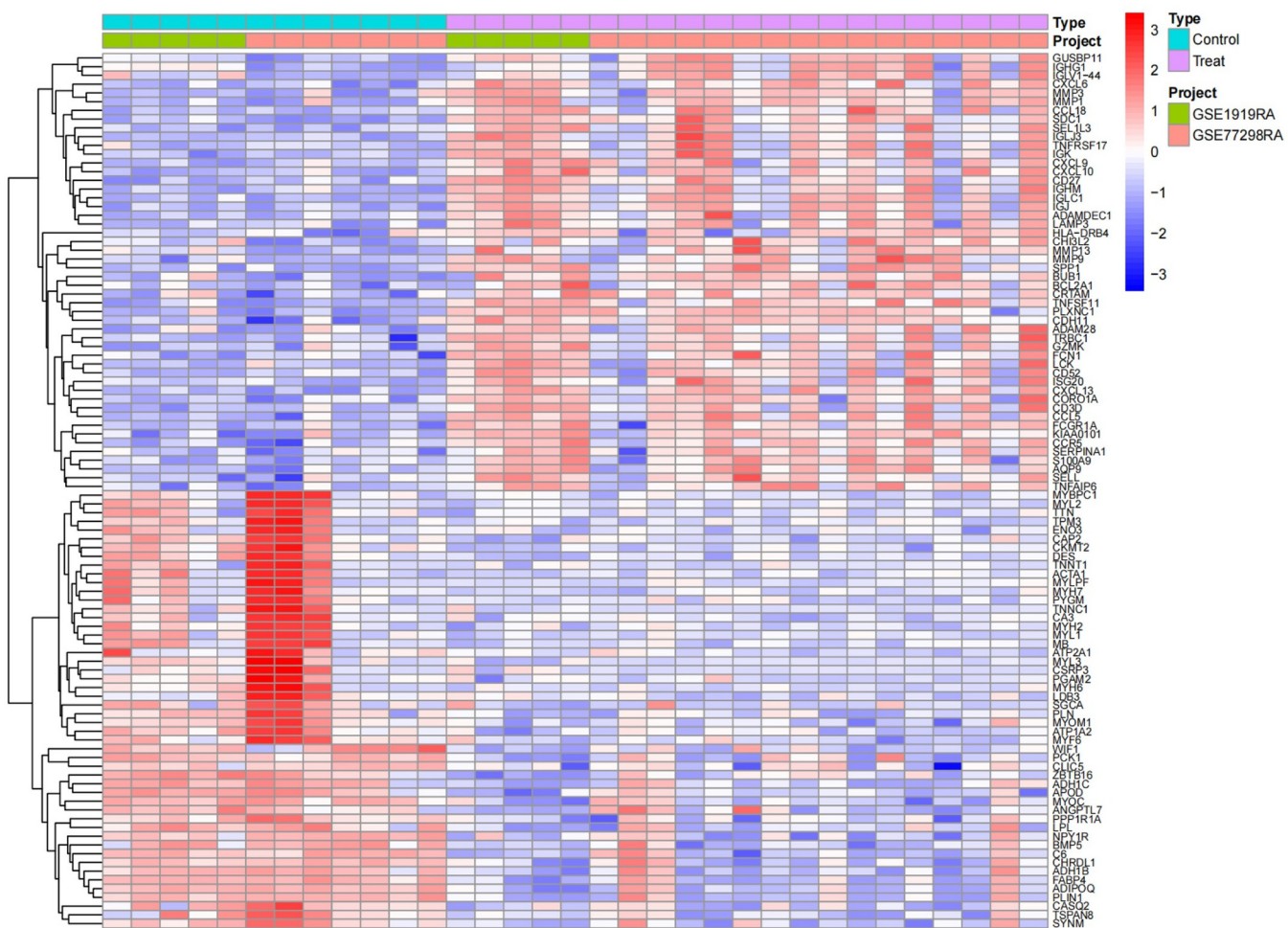

**Fig 12. Heatmap of the RA group, where light blue represents the normal control group, pink represents the disease group, and green and orange differentiate the distinct datasets included, with red signifying high expression and blue representing low expression.**

therapeutic targets for OA and RA. APOD and FASN, being downregulated, might suggest pathways through which their increased expression could ameliorate disease symptoms or progression. In contrast, the upregulation of SDC1 and TNFSF11 points to their roles in disease exacerbation or as key players in the underlying pathogenic mechanisms, presenting them as possible targets for therapeutic intervention aimed at reducing their expression or activity in the disease context(Figs 24–27).

## 9. Transcription factor differential analysis

The differential analysis of transcription factors has unveiled a significant finding regarding Peroxisome proliferator-activated receptor gamma (PPARγ) in relation to Osteoarthritis (OA) and Rheumatoid Arthritis (RA). It has been observed that the expression levels of PPARγ are markedly lower in patients suffering from OA and RA compared to individuals in the normal control group. This discovery is noteworthy, given PPARγ's established role as a pivotal regulator of inflammation and immune responses, which are key components in the pathophysiology of both OA and RA. In RA, the activation of PPARγ has been shown to curb the

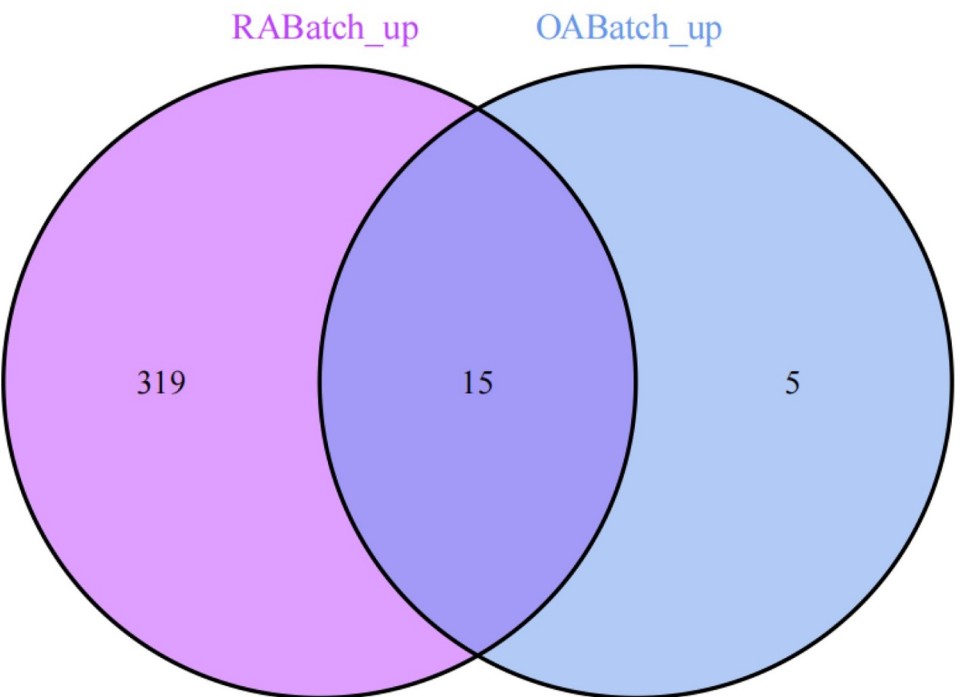

**Fig 13. Shows 15 upregulated common DEGs identified in both OA and RA.**

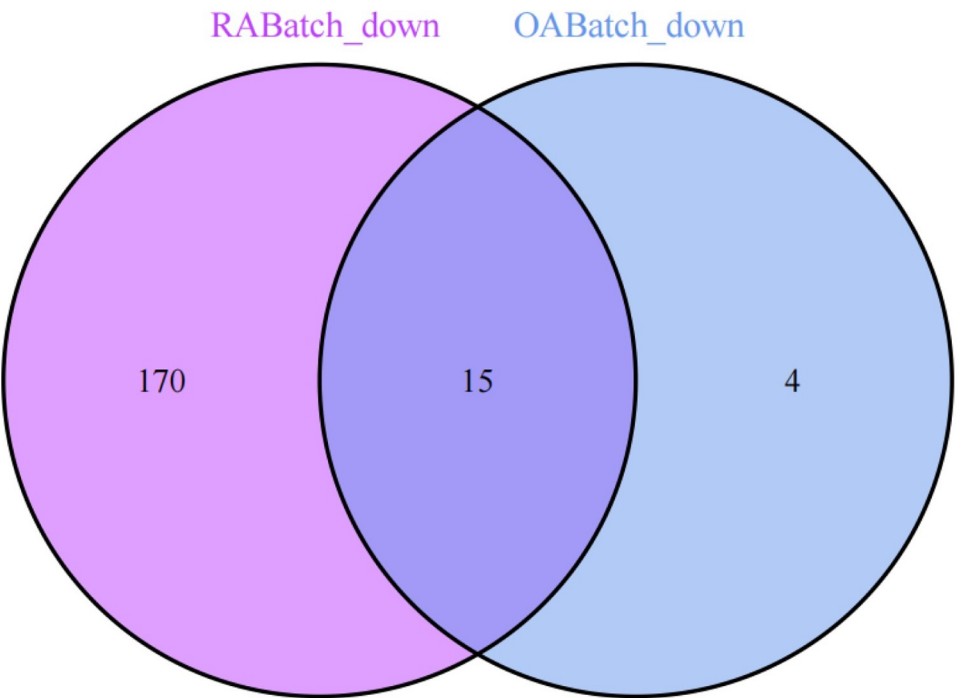

**Fig 14. Displays 15 downregulated common DEGs identified in both OA and RA.**

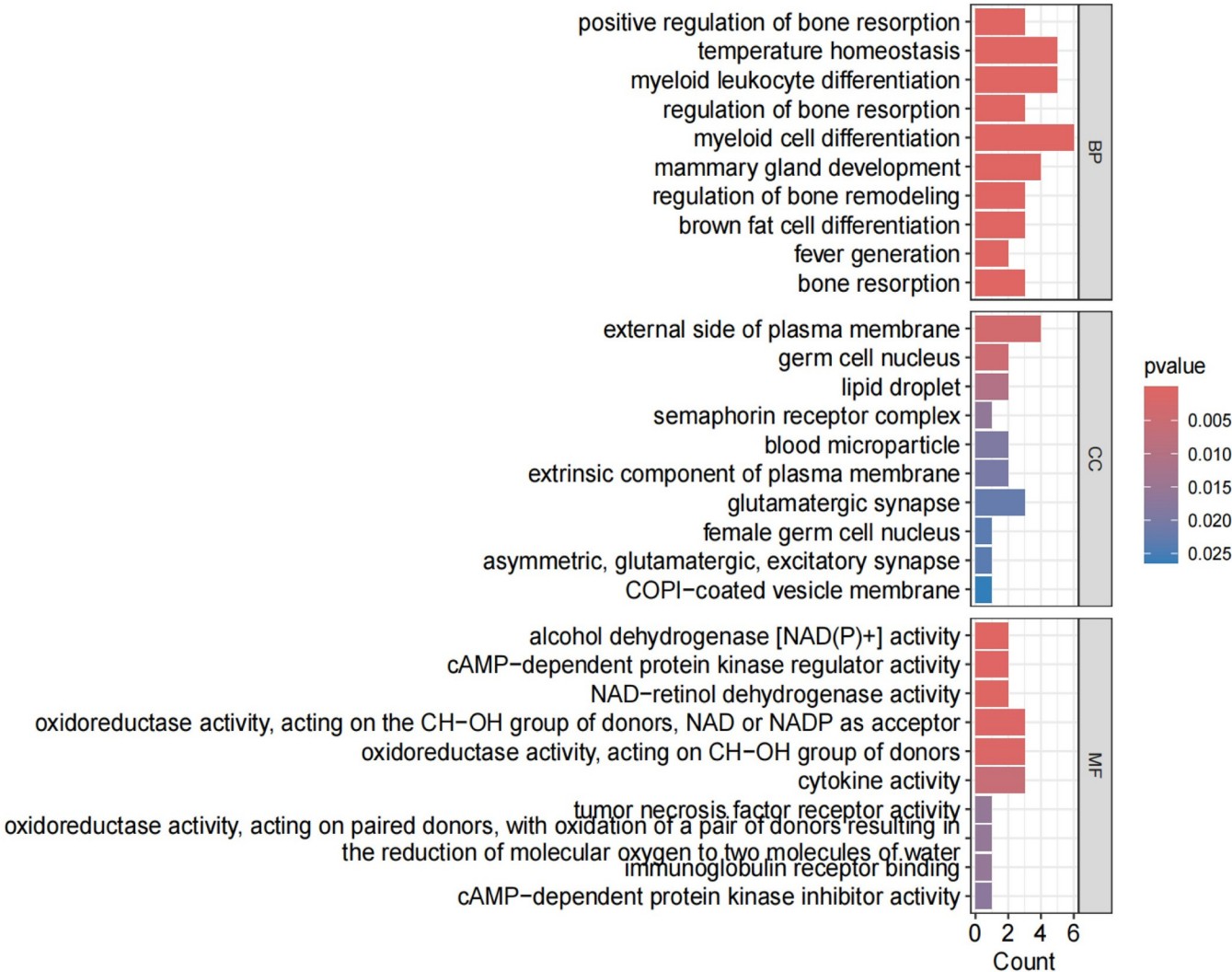

**Fig 15. Shows the GO enrichment analysis pathway diagram, where the x-axis represents the number of enriched genes, and the y-axis represents enrichment relevance.**

proliferation of synovial fibroblasts and reduce the secretion of inflammatory cytokines, underscoring its potential as a therapeutic target to alleviate joint damage. Similarly, in OA, PPARγ's anti-inflammatory properties and its capacity to modulate the joint environment suggest its utility in slowing the disease's progression by preserving joint integrity. Integrating these insights, it becomes evident that the reduced expression of PPARγ in OA and RA patient groups not only illuminates its crucial involvement in the diseases' mechanisms but also marks its significance as a therapeutic target. Given PPARγ's roles in metabolic regulation, inflammation control, and immune response modulation, its potential in arthritis treatment is considerable. The current interest in PPARγ-targeted therapies, especially those initially developed for metabolic conditions like diabetes, signals a promising direction in drug development for arthritis management. This underlines the critical need for continued research to further understand PPARγ's role in arthritis, aiming to innovate therapeutic strategies that can significantly improve outcomes for individuals afflicted with these debilitating conditions(Fig 28).

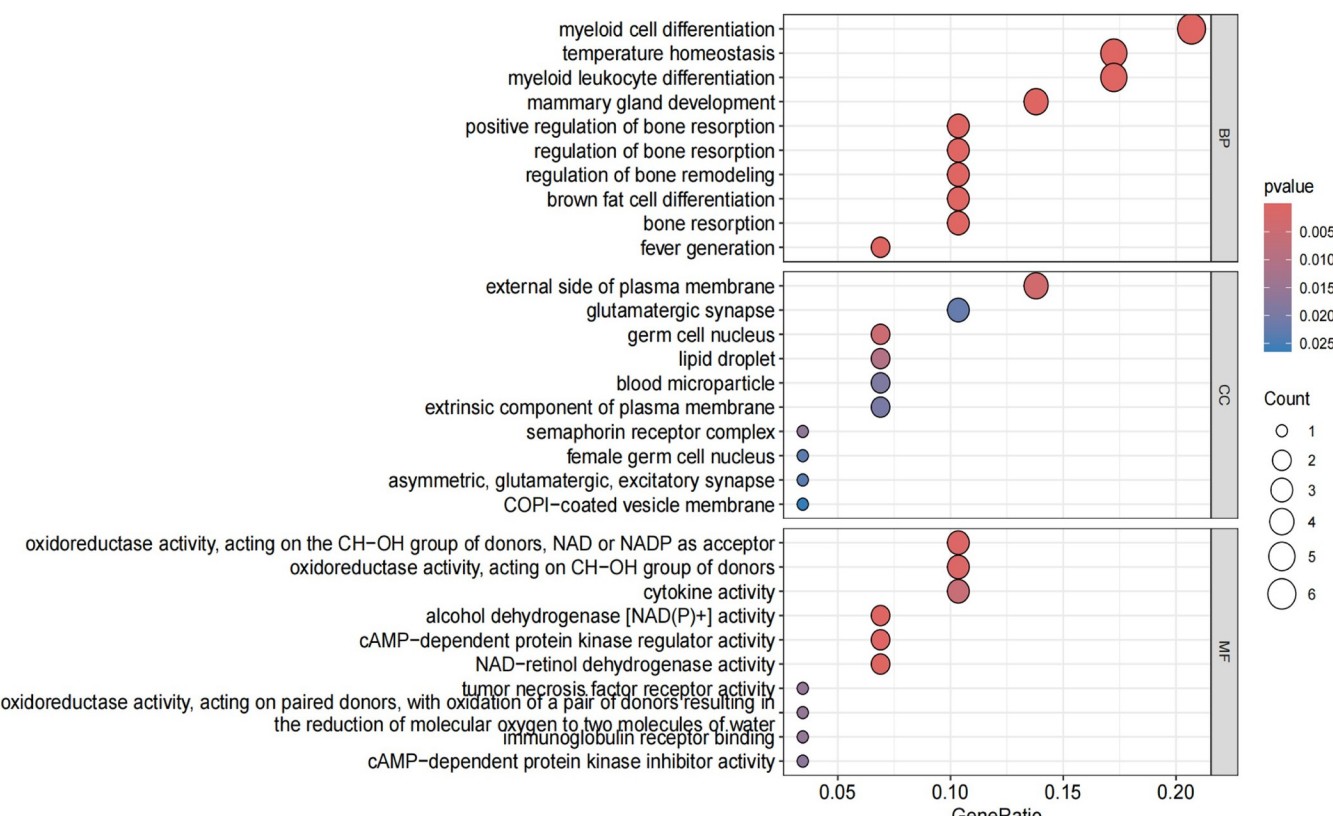

**Fig 16. Displays another aspect of the GO enrichment analysis pathway diagram, with the x-axis indicating the gene ratio and the y-axis showing enrichment relevance.** Red color signifies a positive correlation, while blue color indicates a negative correlation.

## III. Discussion

Rheumatoid arthritis (RA) is a chronic, inflammatory, and autoimmune disease that affects multiple systems in the body [15]. Osteoarthritis (OA), which has traditionally been classified as a non-inflammatory arthritis, is primarily associated with aging and degenerative changes [16]. However, recent advancements in research have uncovered overlapping features in the pathogenesis of both OA and RA. To investigate these potential links, we employed bioinformatics techniques to analyze the expression patterns of specific genes in both conditions. Our analysis identified 10 key genes involved in processes such as temperature stability, heat production induction, and lipid metabolism. These findings suggest that temperature fluctuations and their regulatory mechanisms could be a common risk factor for both diseases, a hypothesis supported by previous research [17–19]. To further validate our initial findings, we conducted an independent analysis of key gene expression, specifically focusing on APOD, FASN, TNFSF11, SDC1, and the transcription factor PPARγ. Our results revealed that the expression levels of APOD and FASN genes, as well as the PPARγ transcription factor, were lower in patients with both OA and RA. Conversely, the expression levels of TNFSF11 and SDC1 genes were higher. These observations align with previous studies and provide deeper insights into the molecular connections and shared pathophysiological features of OA and RA. Our study underscores the significance of these genes in understanding the complex interplay of shared and distinct pathological processes in these autoimmune diseases.

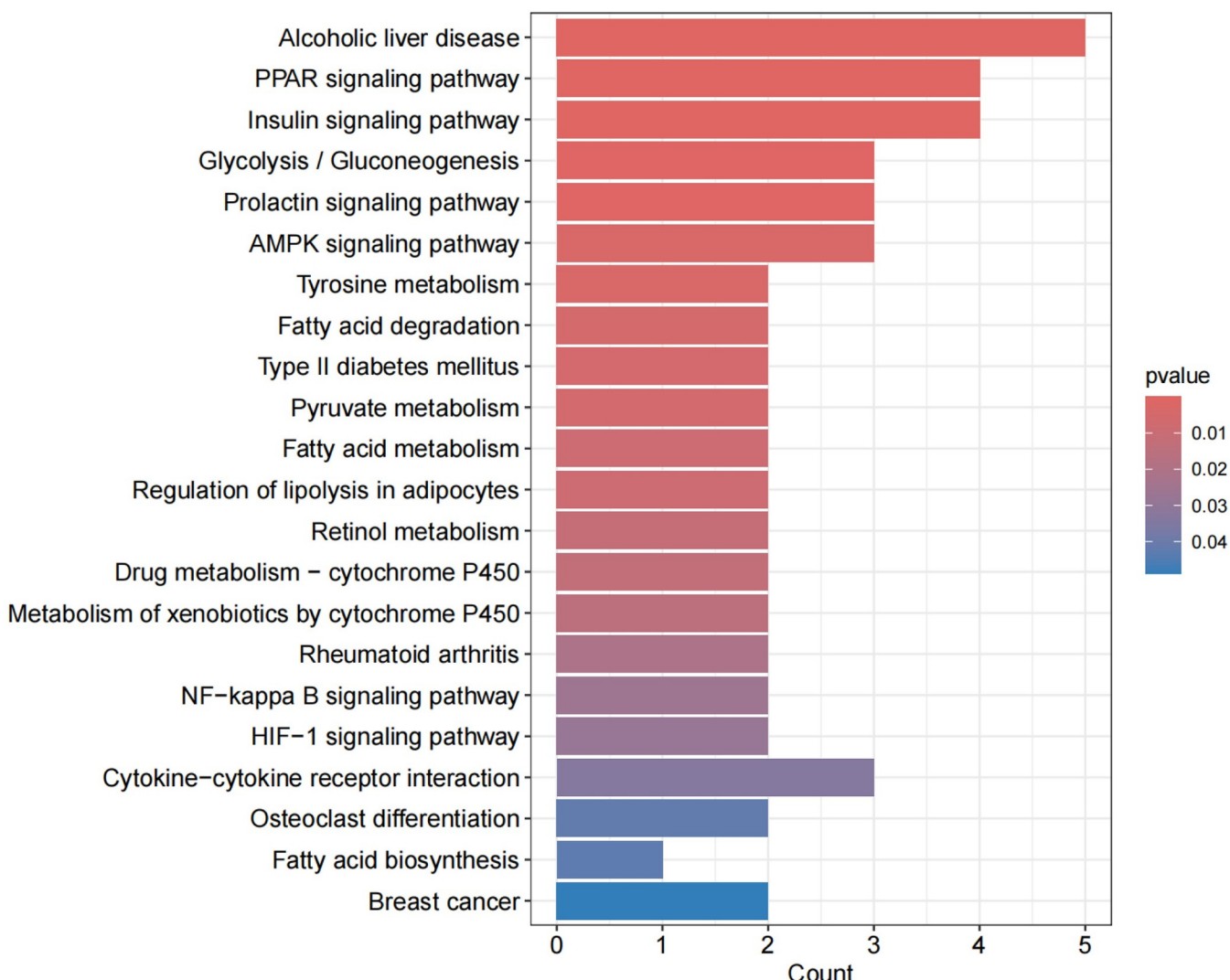

**Fig 17.** Exhibits a KEGG enrichment analysis pathway diagram, where the x-axis displays the number of enriched genes, and the y-axis shows enrichment relevance.

Previous independent studies have revealed the low expression of the differential genes APOD and FASN in single diseases (either OA or RA). For instance, Li et al. (2022) observed significantly lower levels of ApoD in the serum of patients with knee osteoarthritis (KOA) compared to healthy controls [20]. Collins-Racie et al. (2009) found significantly decreased expression levels of LXR target genes ABCG1, as well as apolipoproteins D and E, in OA cartilage [21]. Additionally, in vitro experiments by Xu et al. (2024) demonstrated that APOD delays the progression of OA by influencing the proliferation, apoptosis, and autophagy of FLS and chondrocytes, as well as reducing oxidative stress [22]. Steiner et al. [23] (2008) also showed decreased expression of APOD in RA [23]. In a study by Gong et al. (2023), they discovered that the CircRREB1-FASN axis alleviates the progression of osteoarthritis by inhibiting PI3K-AKT signaling mediated by FGF18 and FGFR3, leading to increased p21 expression, indirectly confirming the downregulation of FASN in OA [24]. Furthermore, He et al. (2020) demonstrated that Bi Zhong Xiao decoction exerts anti-inflammatory effects in RA, primarily by increasing FASN expression and affecting fatty acid metabolism, further confirming the

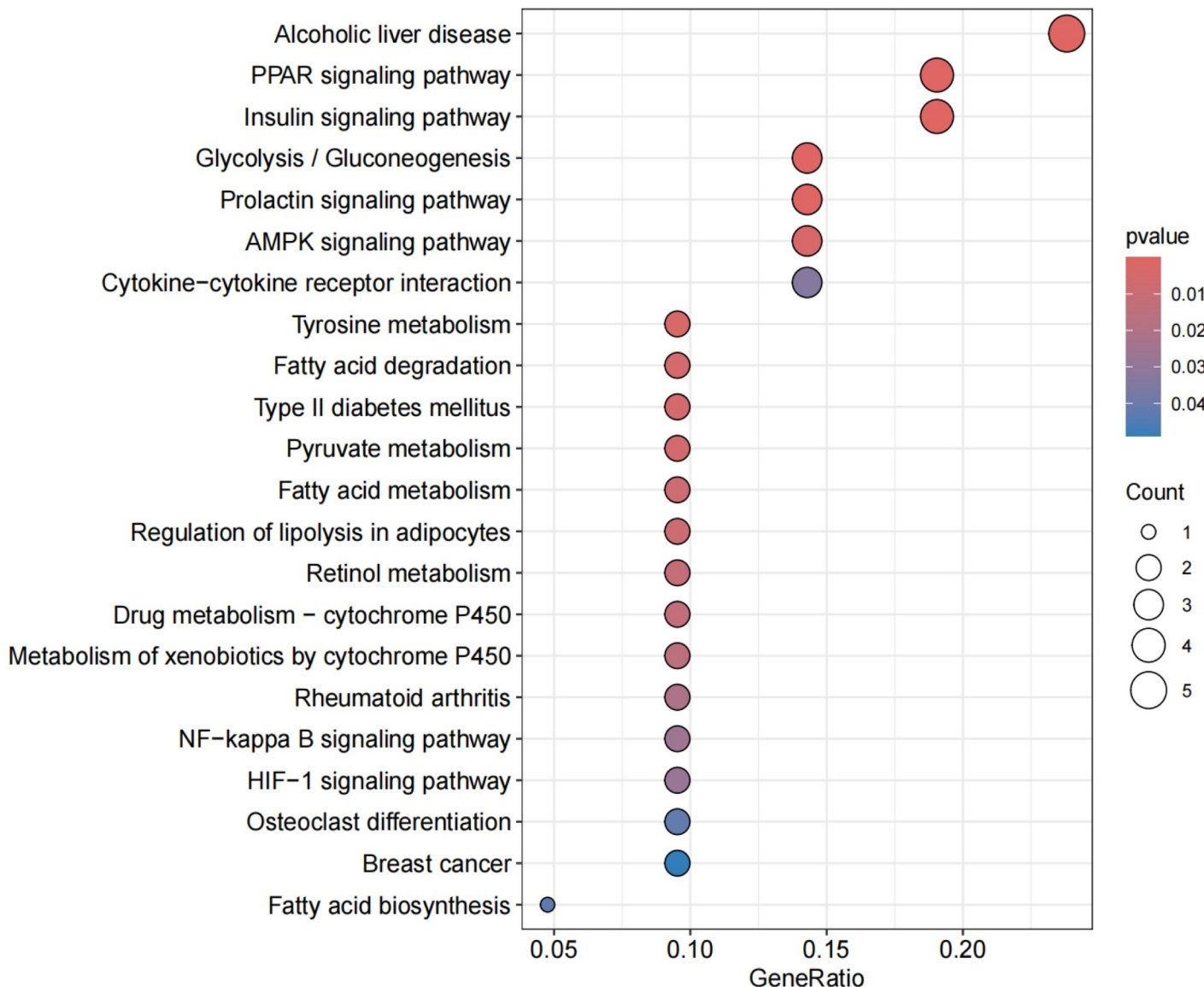

**Fig 18. Demonstrates a KEGG enrichment analysis pathway diagram, with the x-axis indicating the gene ratio and the y-axis denoting enrichment relevance.** Red represents a positive correlation, while blue signifies a negative correlation. These diagrams and charts provide a visual representation of how the common DEGs between OA and RA participate in various biological processes, cellular components, molecular functions, and specific pathways, indicating potential areas of pathophysiological overlap and therapeutic target opportunities for these two conditions.

downregulation of FASN in RA [25].Based on our informatics analysis, we confirmed the low expression of both APOD and FASN in patients with OA and RA. This finding not only extends previous research results but also emphasizes the common low expression pattern of APOD and FASN in OA and RA, providing a new perspective for understanding the shared pathological mechanisms of these two diseases and exploring potential therapeutic strategies. Collectively, these research results reveal the complex interaction between lipid metabolism and inflammatory processes and suggest potential driving forces in the progression of OA and RA. Given the close connection between abnormal lipid metabolism and inflammatory environments, this may serve as an important marker of shared pathological mechanisms in OA and RA. Future research efforts should therefore delve deeper into how APOD and FASN

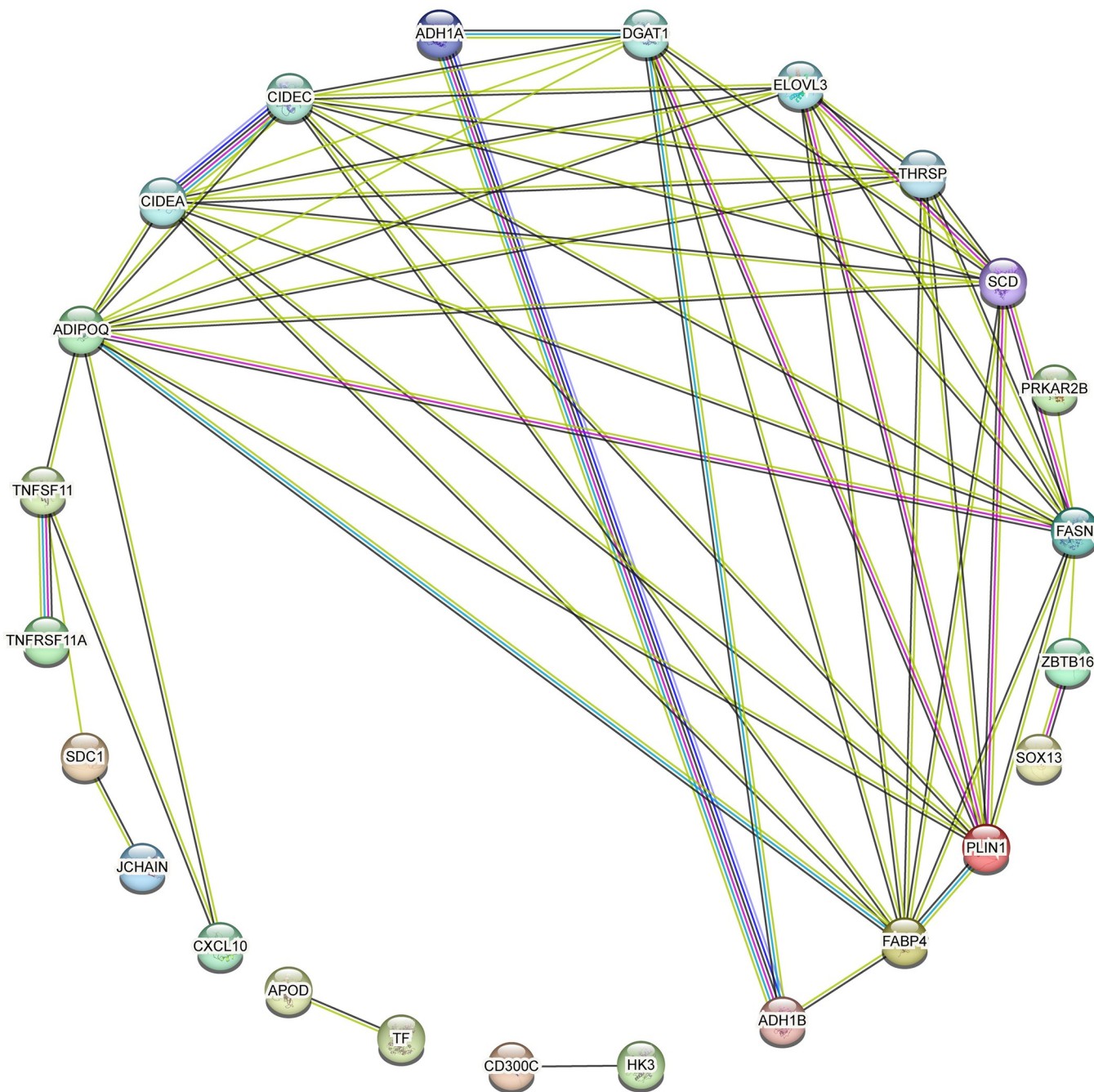

**Fig 19. This part of the Fig would show the protein-protein interaction (PPI) network generated from the 30 commonly expressed differentially expressed genes (DEGs) identified between Osteoarthritis (OA) and Rheumatoid Arthritis (RA).** In this visualization, each node (circle) represents a protein encoded by one of the DEGs, and lines (edges) between nodes represent known or predicted protein-protein interactions. The layout of the network may be designed to emphasize the complexity of interactions, with densely interconnected nodes clustered towards the center.

specifically play roles in inflammation regulation and joint health protection, and how these discoveries can be translated into effective treatment strategies.

Similarly, through an extensive literature review, we have discovered that numerous independent studies have reported the overexpression of TNFSF11 and SDC1 in individual diseases (either OA or RA), revealing their potential roles in disease development. For instance,

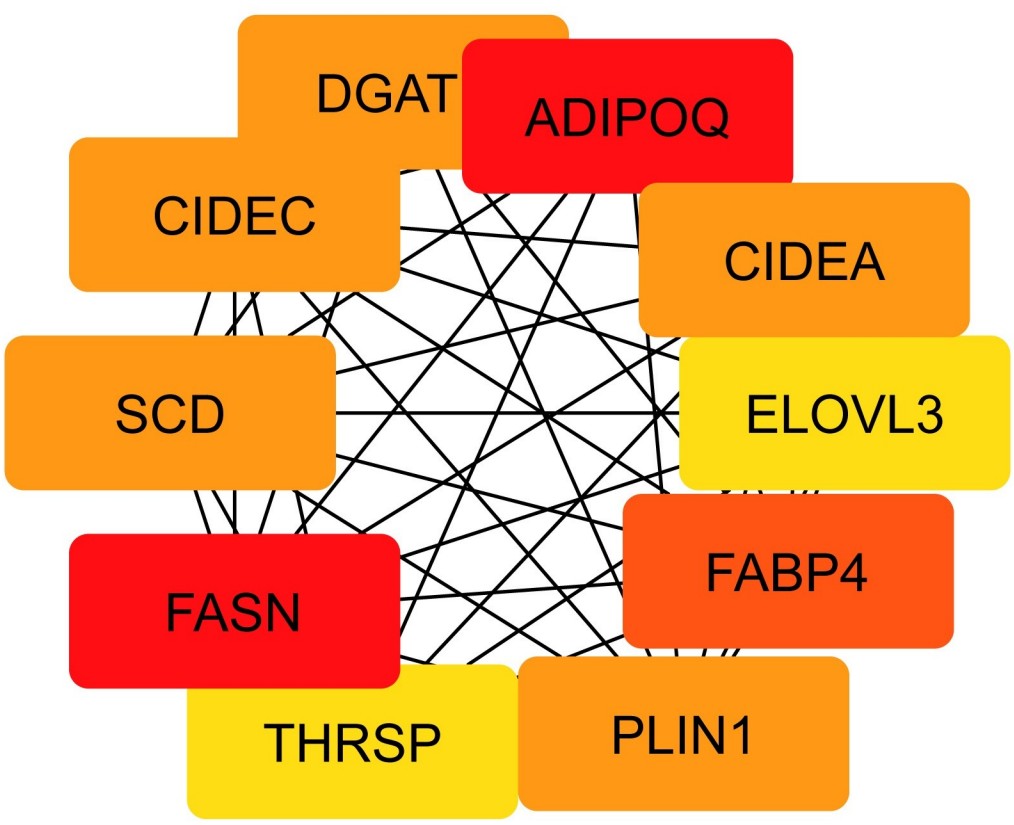

**Fig 20. Focuses on the module comprising the ten identified hub genes: PLIN1, SDC1, CXCL10, SCD, FABP4, FASN, TNFSF11, JCHAIN, APOD, and ADIPOQ.** In this module, the proximity of the nodes to the network center and their color intensity symbolize the level of connectivity and interaction strength with other proteins in the network. Nodes positioned closely to the center in deeper red shades signify a tighter connectivity and a more central role in the networks structure and function. Conversely, nodes depicted in lighter colors and located towards the periphery indicate relatively less connectivity. This visualization underscores the importance of these hub genes in the network and their potential as critical players in the biological processes common to both OA and RA.

Jin et al. (2022) demonstrated how artemisinin alleviates OA progression by reducing TNFSF11 expression in cartilage and inhibiting PI3K/AKT/mTOR signaling, thereby activating mitochondrial autophagy [26]. Additionally, Jiang et al. (2023) found that PD0325901 mitigates RANKL-induced osteoclast formation and cartilage inflammation by suppressing NF-κB and MAPK pathways [27]. In terms of SDC1 research, Wang et al. (2021) elucidated how EZH2 promotes SDC1 expression through histone methylation of the microRNA-138 promoter, leading to cartilage degeneration in OA [28]. Peng et al. (2022) discovered that miR-3960 from mesenchymal stem cell-derived extracellular vesicles inactivates the SDC1/Wnt/β-catenin axis by targeting PHLDA2, thereby alleviating cartilage cell damage in OA [29]. Studies on RA have also shown that multiple research groups reported almost simultaneously an increase in TNFSF11 (RANKL) expression in synovial tissues of RA patients [30–32].Li et al. (2021) pointed out that the small-molecule drug Iguratimod inhibits osteoclastogenesis and bone resorption by mature osteoclasts by interfering with RANKL and TNF-α signaling, demonstrating a bone-protective effect against bone loss-related diseases such as RA, making it a unique option for RA treatment [33]. Furthermore, Osama Alzoubi (2023) proposed that the IL-34/SDC-1 signaling pathway may influence the interaction between bone marrow cells and lymphocytes in RA by regulating glycolysis, thereby promoting the occurrence and

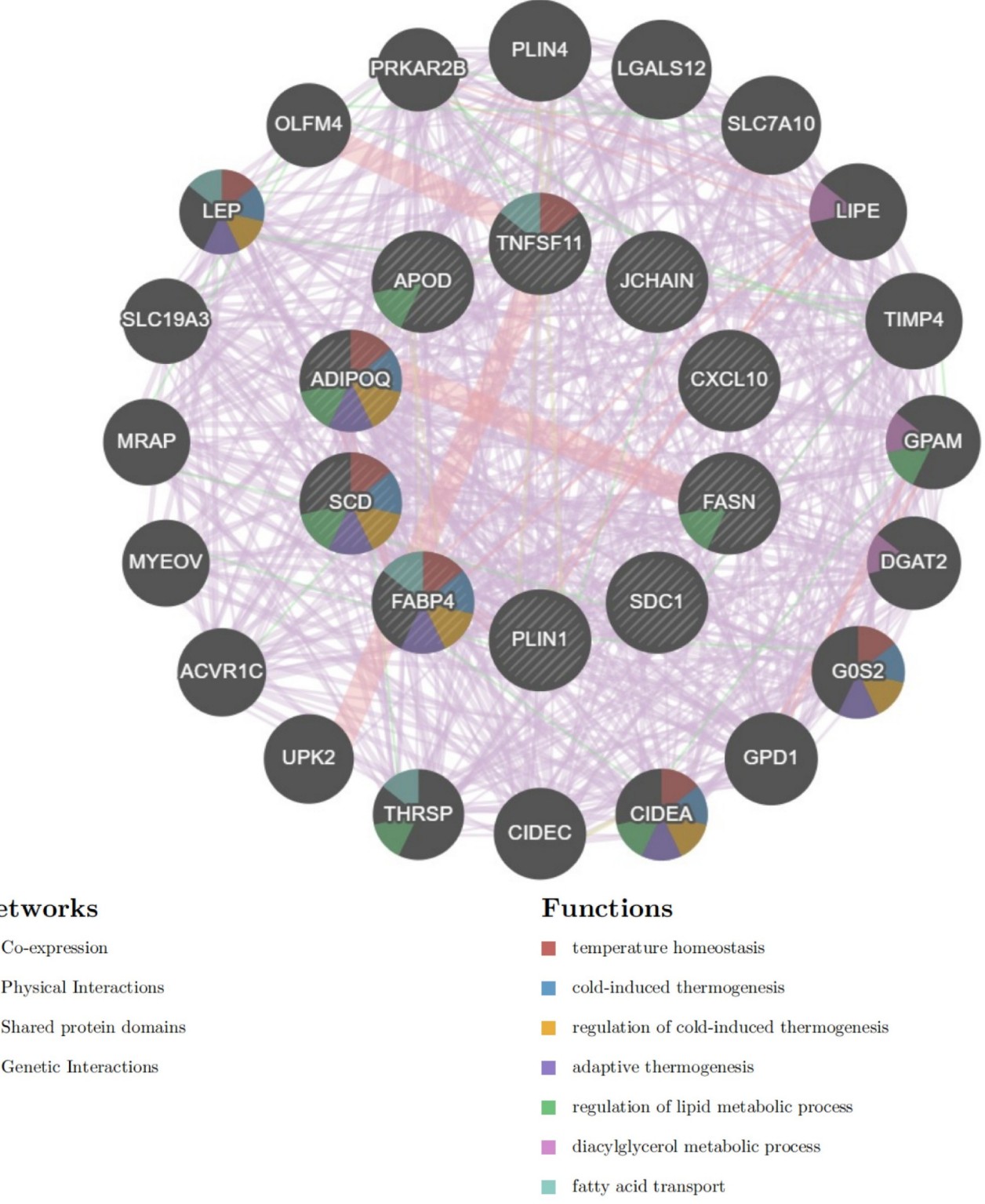

**Fig 21. GENEMANIA analysis visualization for commonly expressed DEGs and their involvement in key biological processes.** This part of the Figure visualizes the involvement of the 10 commonly expressed differentially expressed genes (DEGs), either directly or indirectly through their close connections with the top 20 related genes, in several critical biological processes. These processes include temperature homeostasis, cold-induced thermogenesis, regulation of cold-induced thermogenesis, adaptive thermogenesis, regulation of lipid metabolic process, diacylglycerol metabolic process, and fatty acid transport. The visualization is structured to highlight the complex network of interactions that these genes partake in, which

underlie their role in these essential physiological and metabolic pathways. In this network diagram:Purple Lines indicate co-expression links, illustrating genes that are commonly regulated or exhibit similar expression patterns across various conditions, suggesting potential functional relationships.Red Lines depict physical interactions, indicating that proteins encoded by these genes directly bind to each other, which plays a crucial role in cellular function and signaling pathways.Green Lines signify genetic interactions, where the genetic perturbation (mutation, deletion, overexpression) of one gene affects the phenotype or expression of another, indicating functional relationships that may be critical for understanding disease mechanisms or potential therapeutic targets.Yellow Lines represent shared protein domains, indicating that these proteins have similar structural features which might be crucial for their biological function, interaction with other proteins, or regulation.This comprehensive visualization provides insights into how the identified hub genes and their closely related genes interact within a complex network to regulate essential biological processes relevant to temperature regulation and lipid metabolism.

development of arthritis [34].These findings emphasize the shared roles of TNFSF11 and SDC1 in OA and RA, suggesting that they may be involved in similar pathological processes. Therefore, a deeper understanding of the specific mechanisms of these molecules in inflammation and joint destruction may provide crucial clues for developing novel therapeutic strategies against OA and RA.

Overall, our exploration of the molecular mechanisms underlying both osteoarthritis (OA) and rheumatoid arthritis (RA) has revealed significant similarities between the two diseases. This provides novel insights for the subsequent study of their pathological mechanisms, and valuable perspectives for the development of new treatment strategies and the identification of potential therapeutic targets for OA and RA. Finally, given the limitations of high-throughput omics data in OA and RA, our method should be further validated in larger patient cohorts with OA and RA in future studies.

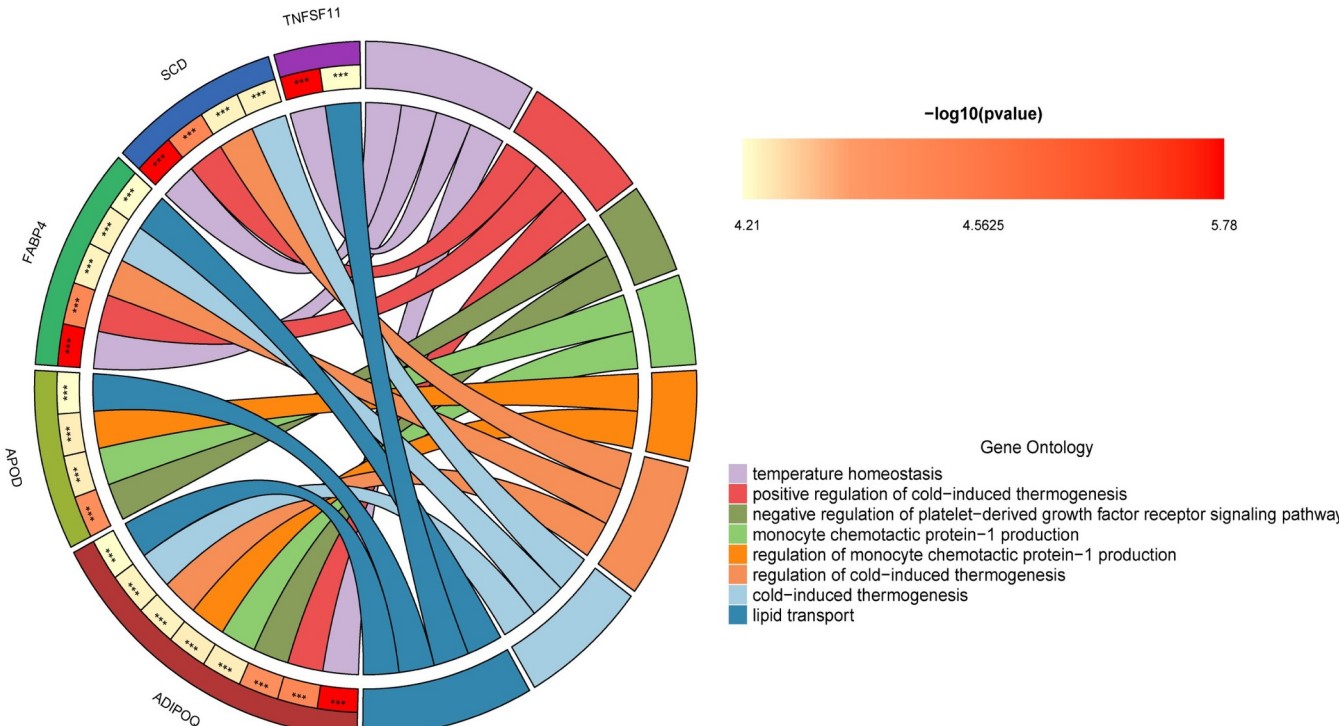

**Fig 22. GO enrichment analysis circular diagram –this diagram represents the results of the gene ontology (GO) enrichment analysis for the hub genes.** The left half-circle's outer edge is color-coded to represent different hub genes, while the right half-circle's outer edge uses different colors to denote various enriched GO pathways. The inner part of the circle uses stars to indicate significance levels of the enrichment: three stars for a p-value < 0.001, indicating highly significant enrichment; two stars for a p-value < 0.01, signifying moderate significance; and one star for a p-value < 0.05, marking general significance in the enrichment analysis.

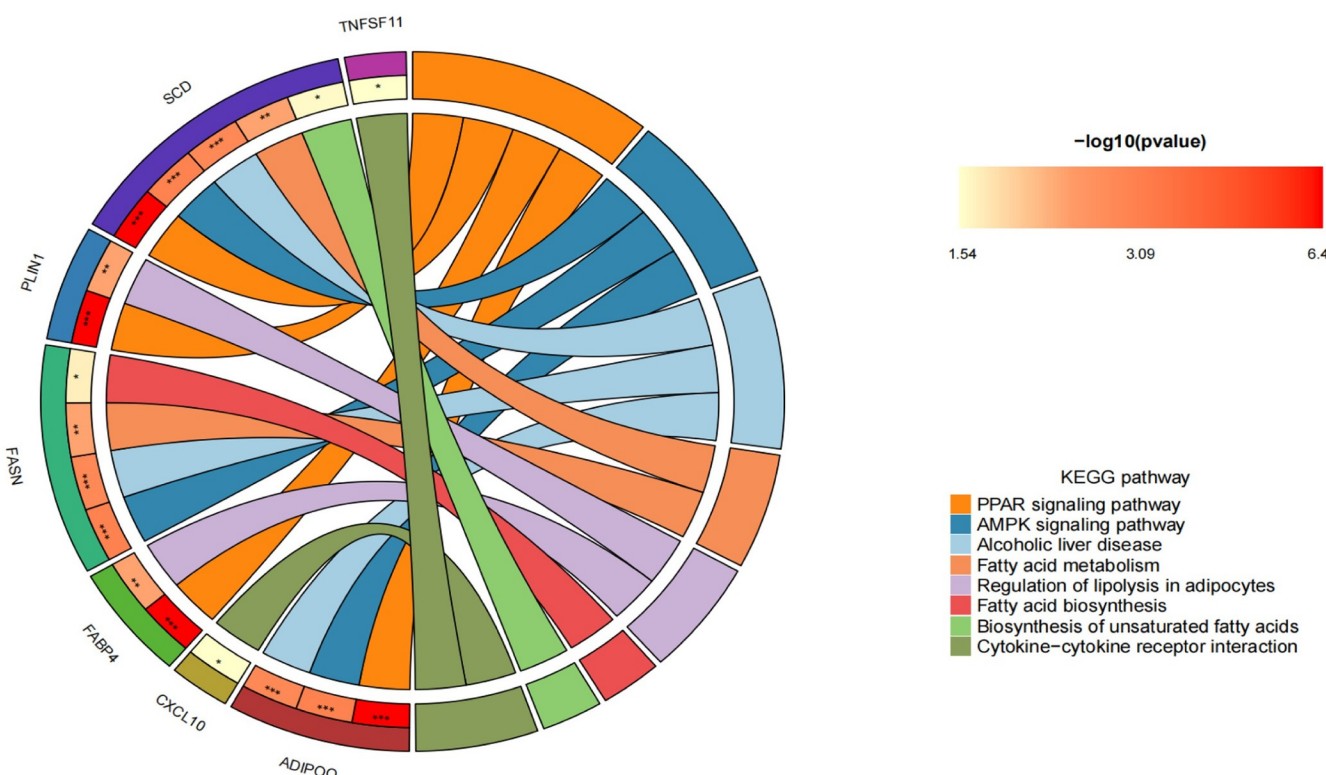

**Fig 23. KEGG enrichment analysis circular diagram –similarly, this diagram displays the kyoto encyclopedia of genes and genomes (KEGG) pathway enrichment analysis results for the hub genes.** The left half-circle's outer edge is color-coded to signify different hub genes while the right half-circle's outer edge uses various colors to represent different enriched KEGG pathways. The significance of the enrichment is denoted by stars in the inner circle, using the same convention as in the GO analysis diagram: three stars for p-value < 0.001, two stars for p-value < 0.01, and one star for p-value < 0.05.

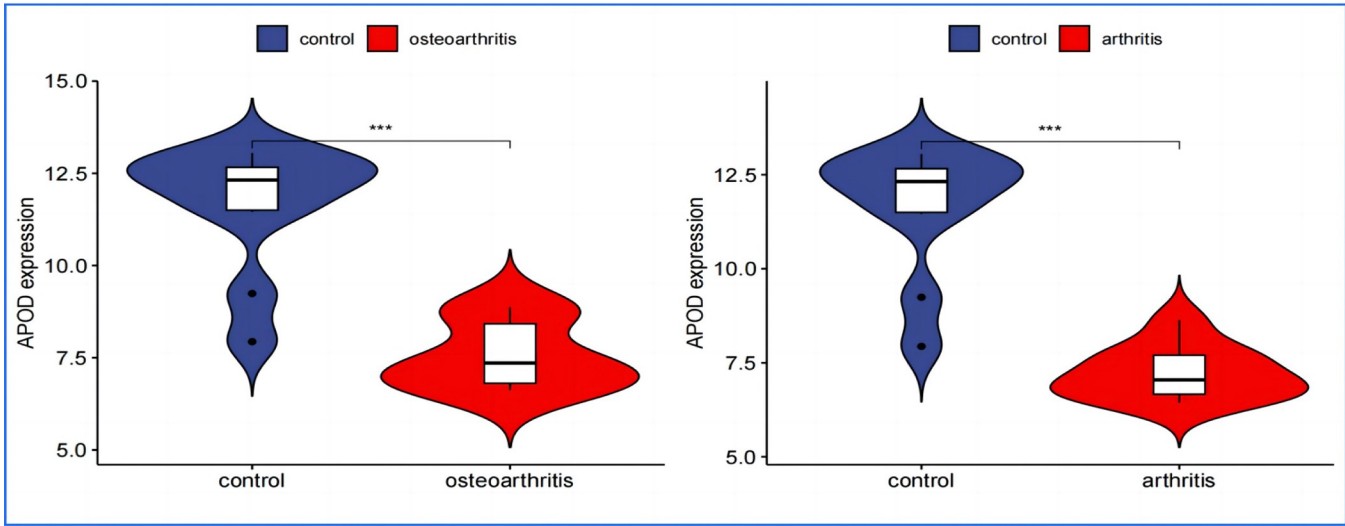

**Fig 24. APOD Expression in OA and RA–illustrates significantly lower expression levels of APOD in both OA and RA disease groups compared to the normal controls, highlighting its potential role in the pathogenesis of these conditions.**

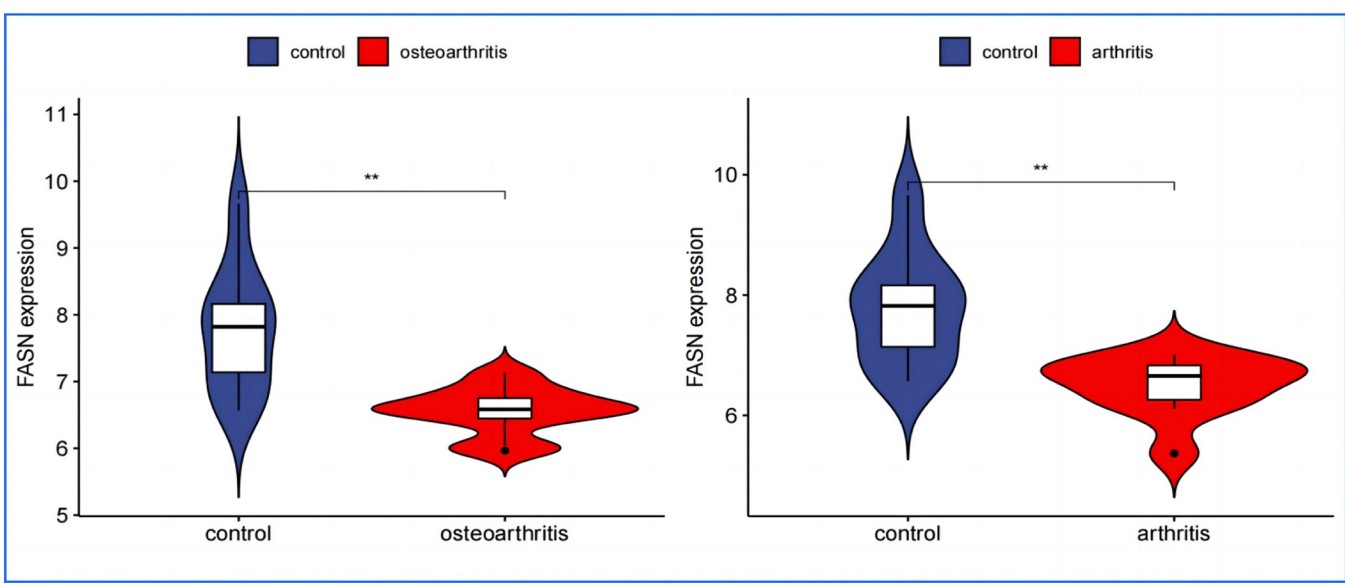

**Fig 25. FASN Expression in OA and RA–shows that FASN is also significantly downregulated in the disease groups for OA and RA, suggesting its decreased activity might influence disease mechanisms or progression.**

## IV. Conclusion

In this study, we successfully identified shared molecular pathways in OA and RA, and screened out potential therapeutic targets. These findings not only validate the existing research foundation, but also significantly enhance our understanding of the pathological mechanisms of these two common diseases. Additionally, they provide new perspectives for developing innovative treatment strategies, potentially improving the therapeutic outcomes and quality of life for patients with these two diseases in the future.

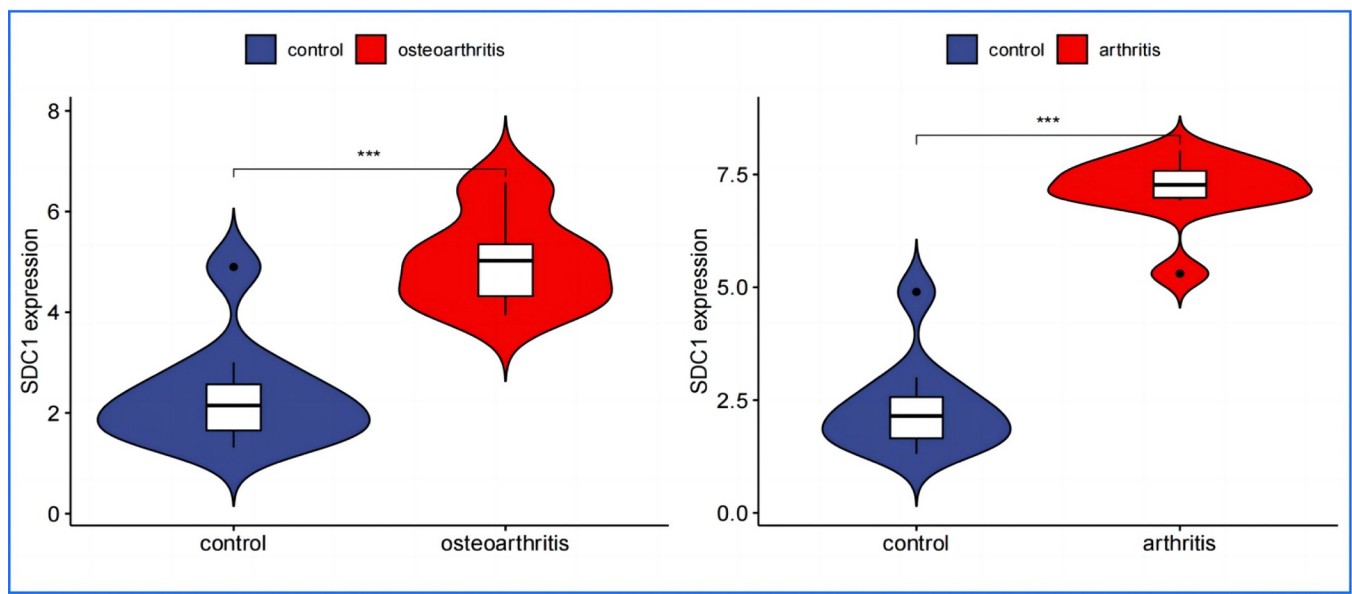

**Fig 26. SDC1 Expression in OA and RA–depicts SDC1 as being significantly upregulated in both OA and RA groups, indicating its heightened activity in the disease context and potential involvement in disease processes.**

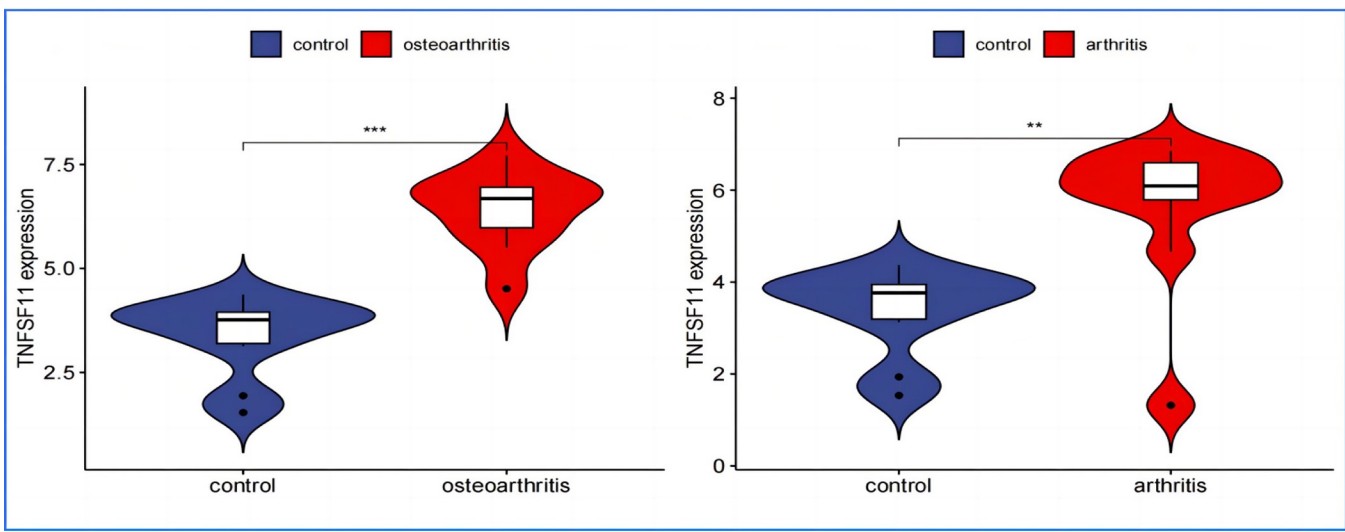

**Fig 27. TNFSF11 Expression in OA and RA–exhibits higher expression levels of TNFSF11 in OA and RA, pointing to its possible contribution to disease pathology or inflammation.**

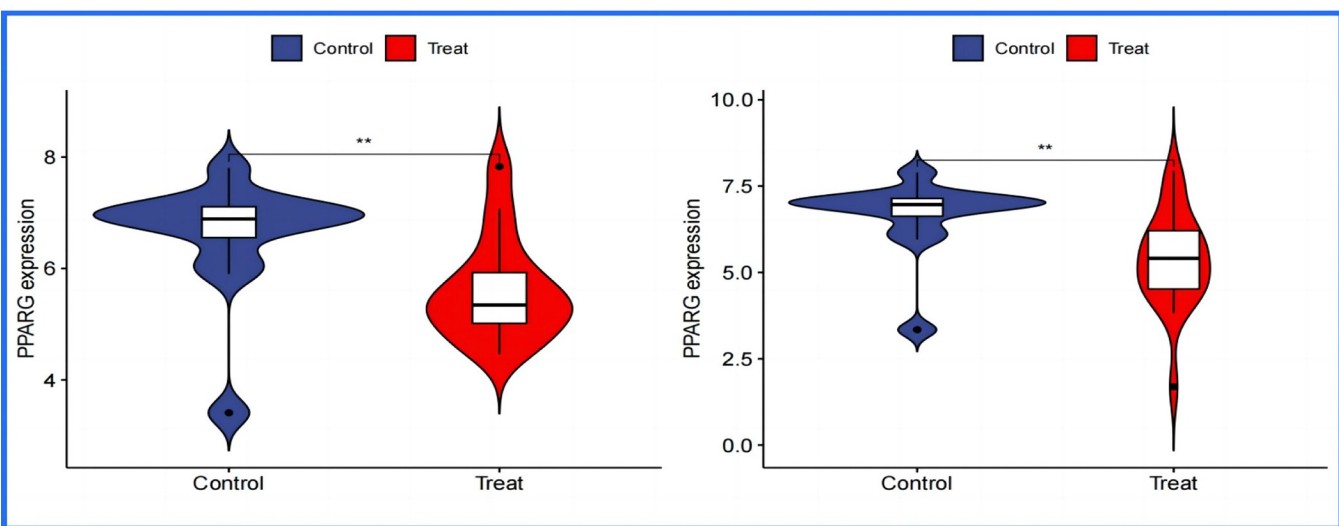

**Fig 28. Differential expression of transcription factor PPARG in OA and RA groups compared to normal control.** Illustrate that the expression of the transcription factor PPARG in both the Osteoarthritis (OA) and Rheumatoid Arthritis (RA) groups is significantly lower than that observed in the normal control group. The horizontal axis uses blue to represent the normal control group, and red to denote the experimental groups (either OA or RA). The vertical axis measures the expression levels of the transcription factor.

## Supporting information

**S1 File.**
(ZIP)

## Author Contributions

**Conceptualization:** Peng-fei Han.

**Data curation:** Chang-sheng Liao, Peng-fei Han.

**Formal analysis:** Chang-sheng Liao, Peng-fei Han.

**Project administration:** Yan Zhang.

**Resources:** Yan Zhang.

**Software:** Xi-yong Li, Yan Zhang.

**Supervision:** Xi-yong Li.

**Validation:** Xi-yong Li.

**Visualization:** Chang-sheng Liao, Fang-zheng He.

**Writing – original draft:** Chang-sheng Liao, Fang-zheng He.

**Writing – review & editing:** Chang-sheng Liao, Fang-zheng He.

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
