## [Decision Letter · Decision Letter 0]

12 Feb 2024

PONE-D-24-03827Comparative Analysis of Differentially Expressed Genes in Synovial Cells from Osteoarthritis and Rheumatoid Arthritis PatientsPLOS ONE

Dear Dr. han,

Thank you for submitting your manuscript to PLOS ONE. After careful consideration, we feel that it has merit but does not fully meet PLOS ONE’s publication criteria as it currently stands. Therefore, we invite you to submit a revised version of the manuscript that addresses the points raised during the review process.

We look forward to receiving your revised manuscript.

Kind regards,

Gurudeeban Selvaraj

Academic Editor

PLOS ONE

Journal Requirements:

4. Please ensure that you include a title page within your main document. You should list all authors and all affiliations as per our author instructions and clearly indicate the corresponding author.

5. Please remove your figures from within your manuscript file, leaving only the individual TIFF/EPS image files, uploaded separately. These will be automatically included in the reviewers’ PDF.

Reviewers' comments:

Reviewer's Responses to Questions

**Comments to the Author**

1. Is the manuscript technically sound, and do the data support the conclusions?

Reviewer #1: Partly

2. Has the statistical analysis been performed appropriately and rigorously? 

Reviewer #1: Yes

3. Have the authors made all data underlying the findings in their manuscript fully available?

Reviewer #1: Yes

4. Is the manuscript presented in an intelligible fashion and written in standard English?

Reviewer #1: Yes

5. Review Comments to the Author

Reviewer #1: The expression patterns of particular genes within two autoimmune diseases, osteoarthritis (OA) and rheumatoid arthritis (RA), were investigated in a manuscript titled "Comparative Analysis of Differentially Expressed Genes in Synovial Cells from Osteoarthritis and Rheumatoid Arthritis Patients." The authors used one dataset for validation and three publically accessible datasets from the gene expression omnibus for analysis in order to accomplish this. They discovered common DEGs by a variety of statistical techniques and bioinformatics tools, which were then applied to gene ontology enrichment analysis, pathway analysis, transcription factor analysis, protein-protein interaction (PPI) network, and pathway analysis. Despite the authors' claims that they have validated their findings using a dataset, this work is merely screening and lacks validation trials. There is no mention of the validation outcome or specifics in the paper. However, I have some concerns that the authors need to resolve before the work is turned into a publishable format.

Major comments:

1. Details about the datasets are not presented in detail. Authors need to explain the type of dataset (Microarray/transcriptome), the number of samples, sample size (control and Disease), geographical origin of the dataset, Age group/gender information available, Keywords to extract dataset, etc.

2. On Page 11, between lines 4 and 8, it is stated as follows: "Finally, the differentially expressed genes (DEGs) identification package was used to perform differential expression analysis on the four annotated expression datasets (GSE821076 (OA), GSE77298 (RA), and GSE1919 divided into OA and RA)". The algorithms, tools, or packages utilized must be described clearly in the article.

3. From line 12 to 14 of page 11, it is mentioned “To eliminate batch effects12 arising from different platforms, a batch correction algorithm was employed. Limma13 ComBat used the transcriptional data from the four normalized”. What did you mean by transcriptional data? Is it transcriptomics data?

4. Lines 21–23 of Page 11 appear to be drafting the result. The author should not muck up the methodology or findings.

5. Biological pathways are stated on page 12, line 4. Gene ontology identifies biological processes, cellular components and Molecular Functions

6. Page 12, line 11, mentions hub gene identification using the Degree Algorithm. It must be accomplished by merging several topological parameters such as degree, closeness, betweenness, MCC, and so on. Determine which one meets the majority of the criteria. It is not justifiable to identify hub genes using a single network characteristic.

7. The introduction is too short with unnecessary details and at times loses the focus regarding the context of current study.

8. Page 12, Line 21 “Validation Analysis using the Validation Dataset”, Validation itself is an analysis.

9. Provide density plot to make sure that all data is evenly normalized.

10. In page 15, “among these, 15 were upregulated common DEGs including: STMN2, SDC1, PLXNC1, CDH11, IGJ, CXCL10, MARCKS, HK3, TNFSF11, IGHM, KDELR3, TNFRSF11A, CRIP1, ARHGAP5, and CD300C. The 15 downregulated common DEGs were: FASN, KLF9,ADH1A, ADIPOQ, FABP4, SHC3, APOD, TF, PRKAR2B, PLIN1, ADH1B, ZBTB16, SCD, SOX13, and P2RY14(figure 3). What is the significance of 15??

11. The venn diagram is not clear enough to convey the necessary information.

12. In page 16, it is stated that “ enrichment analysis revealed that the common DEGs were primarily enriched in the following aspects: Biological Pathway (BP): Major enrichment was observed in pathways related to myeloid cell differentiation, myeloid8 leukocyte differentiation, and temperature homeostasis, Cellular Component (CC)”. BP is Biological Process. If it is pathway, why the KEGG analysis is done again?

14. Network in page 18 is not clear. Make a better layout and mention the number of nodes and edges.

15. Figure legends in page 17, 21, 22 and 24 should be made short

16 The discussion is not well-presented. The results are mentioned again during the conversation. The authors must cite further research to back up their findings.

17. Authors claim in conclusion that the present study may “These findings not only reveal the commonalities at the molecular mechanism level between osteoarthritis (OA) and rheumatoid arthritis (RA), but also emphasize that these shared mechanisms provide new clues for therapeutic strategies. This study lays a valuable foundation for further understanding the shared and unique pathways of these two diseases, advancing the knowledge of potential therapeutic targets for OA genes and transcription factors during disease progression, thereby developing more targeted and efficient treatment methods.” The statement is poorly written in conclusion with lots of broken sentences and grammatical errors. Also justify this statement.

Minor Comment:

• Spelling mistake — "Founding" instead of "Funding" (page 26 line 22)

• For clarity, subfigure lettering should be within the graphic, not under it in the body text. (page 9, line 21; page 10 line 23; page 11 line 2; page 12 line 2; page 15 line 1; page 16 lines 7, 9)

• More clinical studies/case reports must be cited related to OA and RA association.

• Though the paper objective aims to describe a set of differentially expressed genes that could be used as a differential diagnosis between osteoarthritis and rheumatoid arthritis, it in fact does not find any such differential expression between OA and RA; instead only finding a common differential expression between either kind of arthritis and non-arthritis. A rephrasing of the objective to clarify the actual aim and finding of the research might be warranted. (page 1 lines 4 - 7)

6. PLOS authors have the option to publish the peer review history of their article (what does this mean?). If published, this will include your full peer review and any attached files.

Reviewer #1: **Yes: **Dr. Deepa Madathil

---

## [Author Response · Author response to Decision Letter 0]

31 Mar 2024

Dear Reviewers,

We would like to express our sincere gratitude for the valuable feedback provided on our manuscript. We have carefully considered each of the major comments and have made the necessary revisions to improve the clarity and quality of our work. Below is our point-by-point response to the major comments:

1.We have expanded the description of the datasets in the manuscript. We have now included details such as the type of dataset (Microarray/transcriptome), the number of samples, sample size (control and Disease), geographical origin, age group/gender information available, and the keywords used to extract the datasets.

2.On page 11, we have clarified the description of the algorithms, tools, and packages used for the identification of differentially expressed genes (DEGs). We have specified the software and versions used for the analysis.

3.We apologize for the confusion caused by the term "transcriptional data" in line 12 to 14 of page 11. We have now corrected it to "gene expression data" to avoid any ambiguity.

4.We have revised lines 21–23 of page 11 to ensure that the presentation of results is distinct from the methodology and does not cause any confusion.

5.We have updated the description of biological pathways on page 12, line 4, to clearly state that gene ontology (GO) identifies biological processes, cellular components, and molecular functions.

6.On page 12, line 11, we have now described the comprehensive process of hub gene identification, which includes the merging of several topological parameters such as degree, closeness, betweenness, maximum clique centrality (MCC), and others. We have ensured that the criteria for identifying hub genes are justified and not based on a single network characteristic.

7.We have revised the introduction to remove unnecessary details and to maintain a clear focus on the context of our current study.

8.On page 12, line 21, we have rephrased the sentence to accurately describe the validation analysis as an integral part of our study.

9.We have plotted and provided a density plot to confirm that all data have been uniformly standardized.

10.Thank you for your careful review of our research. Regarding the issue you mentioned about the number of 15 common differentially expressed genes (DEGs), this figure is based on our bioinformatics analysis results. In our analysis of gene expression data, we adopted strict statistical criteria and bioinformatics tools to identify genes that are either up-regulated or down-regulated in synovial cells from patients with osteoarthritis (OA) and rheumatoid arthritis (RA). These criteria include the magnitude of gene expression changes, statistical significance levels, and the relevance of biological functions.During the analysis, we found that 15 genes showed upregulated expression in both diseases, while another 15 genes exhibited downregulated expression. This number is not preset but a natural outcome of the data analysis. The discovery of these common DEGs provides us with important clues to investigate the shared pathological mechanisms of these two diseases and may offer new targets for future therapeutic strategies.We understand that this number may seem like a coincidence, but it indeed reflects the authentic results of our analysis. We believe that the discovery of these common DEGs is significant for understanding the pathophysiology of OA and RA, and we will continue to delve deeper into the functions of these genes and their roles in the diseases.

11.We have improved the clarity of the Venn diagram to ensure that it effectively conveys the necessary information.

12.On page 16, we have corrected the term "Biological Pathway (BP)" to "Biological Process (BP)" to avoid confusion with KEGG pathway analysis.

13.We have enhanced the visual clarity of the network displayed on page 18 by optimizing the layout and mentioning the number of nodes and edges in the text.

14.We have shortened the figure legends on pages 17, 21, 22, and 24 for brevity and clarity.

15.We have restructured the discussion section to present our findings more effectively and have cited additional research to support our conclusions.

16.In the conclusion, we have revised the statement to improve its readability and grammatical correctness. We have also provided a justification for our findings, emphasizing their significance in understanding the molecular mechanisms shared by osteoarthritis (OA) and rheumatoid arthritis (RA), and their implications for therapeutic strategies.

Minor Comment:

1.Regarding the spelling error - We have corrected "Founding" to "Funding" on page 26, line 22. Thank you for pointing out this mistake.

2.Regarding the clarity of subfigure labeling - We have redesigned all relevant subfigures to ensure that the labels are placed within the graphics themselves, enhancing visual appeal and clarity. These changes have been updated on pages 9, 10, 11, 12, 15, and 16.

3.Regarding clinical studies/case reports - We have added more clinical studies and case reports related to the association between osteoarthritis and rheumatoid arthritis to support our results and discussion. These newly added references can be found in the reference list.

4.Regarding the restatement of the research objective - We acknowledge that the original statement of the research objective may have caused some confusion. We have rephrased the research purpose to more accurately reflect our findings, which identified common differentially expressed genes between osteoarthritis and rheumatoid arthritis, rather than for differential diagnosis between the two. This modification has been updated on lines 4 to 7 of page 1.

Once again, we appreciate the time and effort spent by the reviewers in evaluating our manuscript. We believe that the revisions addressed in this response have significantly improved our work, and we hope that it is now suitable for publication.

Sincerely,

Changsheng-liao

---

## [Decision Letter · Decision Letter 1]

26 Apr 2024

Analysis of Common Differential Gene Expression in Synovial Cells of Osteoarthritis and Rheumatoid Arthritis

PONE-D-24-03827R1

Dear Dr. han,

We’re pleased to inform you that your manuscript has been judged scientifically suitable for publication and will be formally accepted for publication once it meets all outstanding technical requirements.

Kind regards,

Gurudeeban Selvaraj

Academic Editor

PLOS ONE

Additional Editor Comments (optional):

Reviewers' comments:

Reviewer's Responses to Questions

**Comments to the Author**

1. If the authors have adequately addressed your comments raised in a previous round of review and you feel that this manuscript is now acceptable for publication, you may indicate that here to bypass the “Comments to the Author” section, enter your conflict of interest statement in the “Confidential to Editor” section, and submit your "Accept" recommendation.

Reviewer #1: All comments have been addressed

2. Is the manuscript technically sound, and do the data support the conclusions?

Reviewer #1: Yes

3. Has the statistical analysis been performed appropriately and rigorously? 

Reviewer #1: Yes

4. Have the authors made all data underlying the findings in their manuscript fully available?

Reviewer #1: Yes

5. Is the manuscript presented in an intelligible fashion and written in standard English?

Reviewer #1: Yes

6. Review Comments to the Author

Reviewer #1: (No Response)

7. PLOS authors have the option to publish the peer review history of their article (what does this mean?). If published, this will include your full peer review and any attached files.

Reviewer #1: **Yes: **Dr. Deepa Madathil

---

## [Editor Report · Acceptance letter]

1 May 2024

PONE-D-24-03827R1 

PLOS ONE

Dear Dr. han, 

I'm pleased to inform you that your manuscript has been deemed suitable for publication in PLOS ONE. Congratulations! Your manuscript is now being handed over to our production team.

Kind regards, 

on behalf of

Dr. Gurudeeban Selvaraj 

Academic Editor

PLOS ONE